# Privacy Leakage Prevention in Distilled Datasets: Transforming Initial Private Data

## Abstract

In deep learning for data release scenarios, it is crucial to focus on data privacy protection. Dataset distillation has demonstrated potential in defending against membership inference attacks while maintaining training efficiency. However, this study first identifies that data generated by state-of-the-art dataset distillation methods **strongly resembles** to initial private data, indicating severe privacy leakage. We define this phenomenon as explicit privacy leakage. We theoretically analyze that distilled datasets with a high $IPC^1$ can weaken both the defense against membership inference attacks and explicit privacy. To address this, we propose a plug-and-play module, Kaleidoscopic Transformation (KT), designed to introduce enhanced strong perturbations to the selected real data during the initialization phase while preserving semantic information. Extensive experiments demonstrate that our method ensures both defense against membership inference attacks and explicit privacy, while preserving the generalization performance of the distilled data. Our code will be publicly available.

## 1 Introduction

Deep learning models rely heavily on vast amounts of personal data to train neural networks, making them susceptible to various privacy attacks (Lyu et al., 2020), such as model inversion (Fredrikson et al., 2015), membership inference attacks (Shokri et al., 2017), and property inference attacks (Melis et al., 2019). These vulnerabilities increase the risk of data breaches and misuse. The concerns surrounding data privacy render it impractical for data curators to share their private data and trained models directly, as these vulnerabilities can lead to legal repercussions and heightened security threats (Karale, 2021; Toch et al., 2018). This situation hinders the development and collaboration within the deep learning community. Therefore, providing principled and rigorous privacy protection is essential for sustainable advancement of deep learning research (Fan et al., 2023; Stahl & Wright, 2018; Sharifani & Amini, 2023).

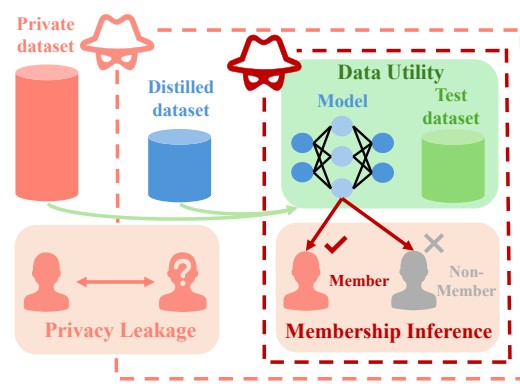

Figure 1: **The release of distilled datasets introduces new privacy risks.** Unlike model-release scenario (dark red box), where attackers can train models for membership inference attacks, the direct access to distilled datasets (pink box) allows for the potential visual exposure of sensitive information if the data is not adequately protected. Meanwhile, the distilled dataset is required to maintain performance comparable to the full private dataset.

In response to these challenges, recent research by Dong et al. (2022) has theoretically and empirically established that *dataset distillation inherently provides a privacy guarantee for models trained on these distilled datasets*. This study highlights the relationship between dataset distillation and differential privacy, demonstrating that models trained on distilled datasets adhere to differential privacy properties. Therefore, releasing models trained on distilled datasets demonstrates potential for defending against membership inference attacks in model-publishing scenarios.

---

[1] $IPC$ means the images per class of the distilled dataset.

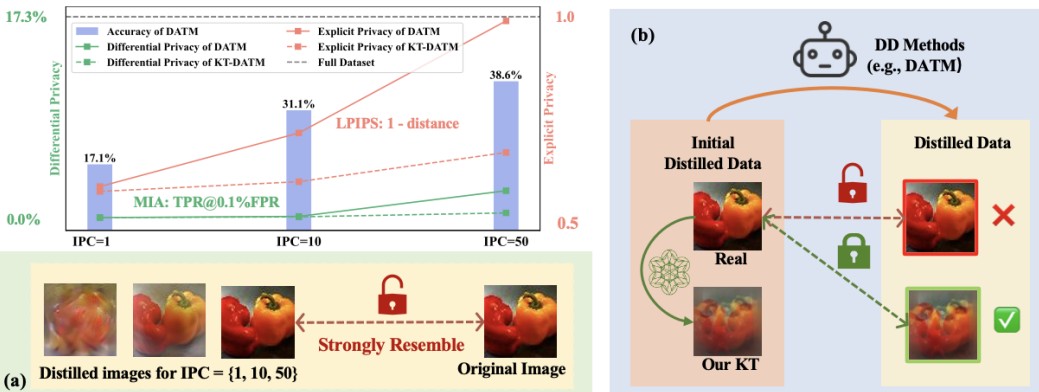

Figure 2: (a) **When IPC ∈ {1, 10, 50}, we examine the differential privacy and explicit privacy leakage, comparing scenarios w/o and w/ our proposed KT.** The below show visualized distilled images corresponding to different IPC values. Differential privacy is assessed via the membership inference attacks using TPR@0.1%FPR (Carlini et al., 2022a), while explicit privacy leakage is evaluated using LPIPS (Zhang et al., 2018). (b) **Overview of the Proposed KT.** As a plug-and-play module, it implements enhanced perturbations to the selected real data at the initialization phase, without participating in the distillation process.

The primary purpose of dataset distillation is to condense large datasets into small generated datasets, where models trained on the distilled data perform comparably to those trained on the original dataset, thereby enhancing training efficiency (Wang et al., 2018; Zhao et al., 2020). Consequently, releasing distilled datasets can enable more people to efficiently train models. Although the distilled data has been verified to adhere to differential privacy properties (Dong et al., 2022), the privacy risks associated with directly releasing data differ from those encountered in the model-release scenario, as illustrated in Figure 1. In the model-release scenario, attackers can only access the model to carry out membership inference attacks (Shokri et al., 2017). In contrast, in the data-release scenario, attackers have direct access to the data itself, allowing them to perform visualizations or train models. If the data is not adequately protected, this could lead to privacy leakage.

As the field of dataset distillation advances, the distilled data generated by the state-of-the-art dataset distillation method e.g. DATM (Guo et al., 2024) *strongly resemble* to the private data, particularly with high IPC (e.g., IPC = 50), as visualized in Figure 2 (a), suggesting severely privacy leakage. We define the phenomenon as **explicit privacy leakage**, characterized by a strong visual similarity between distilled and private images. Furthermore, with a higher IPC, the distilled dataset imposes less stringent privacy restrictions on individual data points, making it more vulnerable to membership inference attacks, as depicted by the solid green line in Figure 2 (a). Consequently, reducing the IPC is necessary to enhance explicit and the defense against membership inference attacks, which inevitably decreases the model performance, as shown by the purple bar in Figure 2 (a).

In this study, we aim to ensure both explicit privacy and resistance to membership inference attacks while maintaining the performance of the distilled dataset. We begin to analyze the sources of privacy leakage in dataset distillation by focusing on two phases: initialization and matching optimization. As demonstrated in Section 3.2, this leakage arises from the common practice of initializing distilled images as real data, a method known for its potential to enhance effectiveness (Dong et al., 2022; Yu et al., 2024). Consequently, we propose a plug-and-play method—Kaleidoscopic Transformation (KT)—aiming at protecting the privacy of selected real data at the initialization phase. KT implements enhanced perturbations on these samples without engaging with the distillation process, thereby being integrated with existing state-of-the-art dataset distillation methods, as illustrated in Figure 2 (b). As a plug-and-play module, with IPC increases, KT ensures both resistance to membership inference attacks (dashed green line) and explicit privacy (dashed red line), as depicted in Figure 2 (a).

**In summary, our contribution is threefold:**

(a) To the best of our knowledge, this work is the first to explore the explicit privacy of the distilled data. We reveal that when IPC is high, the distilled images *strongly resemble* to the initial private images, indicating a significant explicit privacy leakage.

(b) Through theoretical analysis of multiple phases in dataset distillation, we identify that random training sample initialization is the root cause of explicit privacy leakage, as subsequent matching perturbations provide insufficient protection.

(c) Building on these insights, we propose a plug-and-play module, Kaleidoscopic Transformation (KT), to implement enhanced perturbations to the selected real data at the initialization phase. Extensive experiments show that KT effectively protects explicit privacy and defends against membership inference attacks while maintaining generalization performance.

# 2 RELATED WORK

The scenario of data-release introduces new privacy risks compared to model-release scenarios. In traditional machine learning, published models are often targeted by privacy attacks. Common defenses include differential privacy (Dwork et al., 2006). However, in data-release scenarios, we must ensure the published data itself is privacy-protected. Attackers can directly access the data for visualization or model training. Thus, we need to integrate protection into data generation, using methods like Data Generators (Goodfellow et al., 2014) and Dataset Distillation (Wang et al., 2018).

## 2.1 MODEL-CENTRIC METHODS FOR PRIVACY PRESERVATION

**Differential Privacy.** Differential Privacy (Dwork et al., 2006) is a privacy-preserving technique that introduces perturbation into the outputs to obfuscate the accurate return value, thereby preventing the adversary from learning the exact private information (Dwork et al., 2006; Farayola et al., 2024). Shokri et al. (2017) first indicate that the learning task based on differential privacy can reduce the success probability of the membership inference attack against this task. Jayaraman & Evans (2019) evaluate the effectiveness of $(\epsilon, \sigma)$-DP and its variants in neural network models by using membership inference attack. The application of differential privacy spans various domains, including health (Torfi et al., 2022; Adnan et al., 2022), as well as finance (Wang et al., 2022b).

## 2.2 DATA-CENTRIC METHODS FOR PRIVACY PRESERVATION

**Data Generator.** Generative models can serve as an alternative for data sharing (Goodfellow et al., 2014). However, Chen et al. (2020) demonstrate that privacy risks exist not only when training with raw data but also when using synthetic data produced by these generative models. To address this issue, researchers have applied differential privacy (DP) (Dwork et al., 2006) to develop differentially private data generators (referred to as DP-generators) (Xie et al., 2018; Cao et al., 2021; Harder et al., 2021; Ghalebikesabi et al., 2023). However, the noise introduced by differential privacy often results in low-quality generated data, which impedes its effectiveness. Additionally, the training of DP-generators can incur significant computational costs.

**Dataset Distillation.** Dataset distillation (Wang et al., 2018) aims to improve training efficiency by extracting knowledge from a large-scale dataset and construct a significantly smaller distilled dataset, enabling models trained on it achieve comparable performance to those trained on original dataset. Current solutions can be categorized based on their optimization mechanisms (Lei & Tao, 2023), including Gradient Matching (GM) (Zhao et al., 2020; Zhao & Bilen, 2021; Kim et al., 2022), Distribution Matching (DM) (Zhao & Bilen, 2023; Yin et al., 2023), Trajectory Matching (TM) (Cazenavette et al., 2022; Guo et al., 2024). Remarkably, RDED (Sun et al., 2024) introduces an optimization-free paradigm, which directly crop and select realistic patches from the original data, and then stitch the selected patches into the new images as the distilled dataset. It achieves promising performance, particularly for large-scale and high-resolution datasets.

As the field progresses, state-of-the-art dataset distillation methods (Yin et al., 2023; Guo et al., 2024; Sun et al., 2024) are able to produce distilled data that achieve performance comparable to the original data. However, these distilled data closely resemble to real private data, especially at high `IPC` (e.g., `IPC` = 50). *This strong resemblance raises significant privacy concerns, necessitating urgent measures to safeguard the privacy of the distilled datasets.*

**Privacy of Distilled dataset.** Dong et al. (2022) first build the connection between dataset distillation and differential privacy, proving that distilled data—generated via DM (Zhao & Bilen, 2023), DSA (Zhao & Bilen, 2021), and KIP (Nguyen et al., 2020)—can satisfy the definition of differential privacy. However, Carlini et al. (2022b) point out that Dong et al. (2022) incorrectly used Assumption 4.8, thus failing to provide privacy guarantees. Furthermore, recent state-of-the-art dataset distillation

methods, including TM-based methods, such as MTT (Cazenavette et al., 2022), DATM (Guo et al., 2024) and non-optimization-based methods like RDED (Sun et al., 2024), have not been considered. Therefore, we focuses on examining the privacy of distilled datasets generated by these state-of-the-art distillation methods, from both theoretical and empirical perspectives in Section 3.2 and Section 4 .

# 3 PRIVACY ANALYSIS AND PROTECTION IN DATASET DISTILLATION

This section begins by introducing preliminary definitions. Subsequently, we theoretically demonstrate that the distilled dataset with high IPC weakens differential privacy preservation and also causes severely explicit privacy leverage. Our analysis reveals that the issues predominantly arises from the common practice of initializing distilled imaegs as real data. To address these challenges, we propose a plug-and-play module, named KT, which applies expanded transformations to the selected real samples during initialization. KT ensures both differential privacy and explicit privacy while maintaining the generalization performance of the distilled data.

## 3.1 PRELIMINARY

**Dataset distillation.** Given a large-scale dataset $\mathcal{T} = \{\mathbf{x}_i, y_i\}_{i=1}^{|\mathcal{T}|}$, where $\mathbf{x}_i \in \mathbb{R}^d$ is the input sample and $y_i \in \{1, \ldots, C\}$ is the corresponding label, dataset distillation (Wang et al., 2018) aims to synthesize a smaller distilled dataset $\mathcal{S} = \{\tilde{\mathbf{x}}_j, \tilde{y}_j\}_{j=1}^{|\mathcal{S}|}$ with $|\mathcal{S}|$ synthetic samples (i.e., $|\mathcal{S}| \ll |\mathcal{T}|$) such that models trained on $\mathcal{S}$ will have similar test performance as models trained on $\mathcal{T}$:

$$\mathbb{E}_{(\mathbf{x},y) \sim P_D} \left[ \ell \left( \phi_{\boldsymbol{\theta}_{\mathcal{T}}}(\mathbf{x}), y \right) \right] \simeq \mathbb{E}_{(\mathbf{x},y) \sim P_D} \left[ \ell \left( \phi_{\boldsymbol{\theta}_{\mathcal{S}}}(\mathbf{x}), y \right) \right] , \tag{1}$$

where $P_D$ is the test real distribution, $\mathbf{x}$ is a data sample, $\ell$ is the loss function (e.g., cross-entropy loss), and $\boldsymbol{\theta}_{\mathcal{T}}$ and $\boldsymbol{\theta}_{\mathcal{S}}$ denote the parameters of the neural network $\phi$ trained on $\mathcal{T}$ and $\mathcal{S}$, respectively.

In this paper, we decompose the dataset distillation process into two phases: initialization of the distilled data and the subsequent matching optimization, based on a review of previous studies (Guo et al., 2024). The first phase involves the initialization of distilled data, where the common strategy is to utilize real data (Yin et al., 2023; Guo et al., 2024; Sun et al., 2024). The second phase focuses on optimizing this distilled data via various matching mechanisms, as elaborated in Section 2.2 .

**Privacy attack.** Following prior research (Dong et al., 2022; Carlini et al., 2022b), this work mainly focus on membership inference, as it is the most widely studied privacy attack (Hu et al., 2022; 2023; Niu et al., 2024). These attacks aim to determine whether a specific data point was used in training, directly impacting individual privacy.

Moreover, we conduct experiment using the state-of-the-art Likelihood Ratio Attack (LiRA) (Carlini et al., 2022a) because of its high attack performance. LiRA utilizes multiple queries with various data transformations to mitigate the potential privacy-enhancing effects of data augmentation techniques. This approach ensures a more robust evaluation of privacy risks in the context of distilled datasets. A detailed description of the LiRA is provided in Appendix A.2 .

**Differential privacy.** Differential privacy (Dwork et al., 2006) introduces perturbation into the outputs to obfuscate the accurate return value, quantifying and limiting the exposure of individual information. If a mechanism can achieve differential privacy, it can be defined as follows:

> **Definition 1 (Differential privacy) .** *A randomized mechanism $\mathcal{M}$ with range $\mathcal{R}$ is $(\epsilon, \delta)$-DP, if for any two neighboring datasets $D$ and $D'$ which differ in exactly one element, and for any subset $\mathcal{O}$ of possible outputs of $\mathcal{M}$, the following holds:*
>
> $$\Pr[\mathcal{M}(D) \in \mathcal{O}] \leq e^\epsilon \cdot \Pr[\mathcal{M}(D') \in \mathcal{O}] + \delta . \tag{2}$$

**Explicit Privacy.** As our first contribution, we introduce the concept of *explicit privacy*. Explicit privacy refers to the visual similarity between a distilled dataset and the real data used for initialization, reflecting the level of privacy protection at the data level, as shown in Figure 2 (a). It quantifies the risk of directly observable privacy leakage in the resulting data after the distillation process, distinct from the model-level privacy concepts in traditional machine learning (Papernot et al., 2016; Kong & Munoz Medina, 2024).

> **Definition 2 (Explicit privacy) .** *For a distilled dataset $\mathcal{S}$ and a real dataset $\mathcal{T}$, explicit privacy is protected if the following condition is satisfied:*
>
> $$E(\mathcal{S}, \mathcal{T}) = \frac{1}{|\mathcal{S}|} \sum_{\mathbf{x}_\mathcal{S} \in \mathcal{S}} \min_{\mathbf{x}_\mathcal{T} \in \mathcal{T}} Sim(\mathbf{x}_\mathcal{S}, \mathbf{x}_\mathcal{T}) < \tau \,, \tag{3}$$
>
> *where $E(\mathcal{S}, \mathcal{T})$ is the average minimum similarity between any two samples in $\mathcal{S}$ and $\mathcal{T}$, $Sim(\mathbf{x}_\mathcal{S}, \mathbf{x}_\mathcal{T})$ is the similarity between two samples $\mathbf{x}_\mathcal{S}$ and $\mathbf{x}_\mathcal{T}$, and $\tau$ is the threshold.*

We employ the Learned Perceptual Image Patch Similarity (LPIPS) (Zhang et al., 2018) to quantify similarity, thus $Sim(\mathbf{x}_\mathcal{S}, \mathbf{x}_\mathcal{T}) = 1 - \text{LPIPS}(\mathbf{x}_\mathcal{S}, \mathbf{x}_\mathcal{T})$. Unlike pixel-based metrics (Wang et al., 2004; Zhang et al., 2011), LPIPS captures the perceptual differences that are more relevant to privacy concerns in the distilled datasets.

### 3.2 Privacy Bound of Models Trained on Distilled Data

Following Dong et al. (2022), we begin by studying the privacy bound of models trained on distilled data in a differential privacy (DP) manner: *how does removing one sample in the original dataset impact models trained on distilled dataset.* It is important to highlight that our demonstration diverges from that of Dong et al. (2022) because we avoid the non-rigorous assumption in Dong et al. (2022). Our analysis focuses on the two phases of dataset distillation: the initialization of the distilled data and the subsequent matching optimization. We individually assess the differential privacy property of each phase, as elaborated in Proposition 1 and Theorem 1 .

**Phase 1: Differential privacy brought by random sampling initialization is unreliable.** To enhance the performance of distilled datasets, most dataset distillation methods use random sampling from real data as the initialization for distilled data (Sun et al., 2024; Guo et al., 2024; Yin et al., 2023). Therefore, we analyze the differential privacy guarantees of this initialization method using the following proposition.

> **Proposition 1 .** *Given a training dataset of size $|\mathcal{T}|$, random sampling without replacement achieves $(\ln \frac{|\mathcal{T}|+1}{|\mathcal{T}|+1-|\mathcal{S}|}, \frac{|\mathcal{S}|}{|\mathcal{T}|})$-differential privacy, where $|\mathcal{S}|$ is the subsample size.*

This proposition suggests that random sampling initialization achieves differential privacy through randomized response (Dwork et al., 2014). (See Appendix B for proof details.)

However, $\delta = |\mathcal{S}|/|\mathcal{T}|$ reflects the leakage of private data used for initialization and is proportional to IPC. If subsequent distillation phases fail to introduce sufficient randomness to the initialized distilled dataset, this could directly expose the training data used for initialization, motivating us to introduce the concept of explicit privacy leakage.

**Phase 2: The volatility of the matching optimization introduces additional randomness to the distilled dataset, limiting individual data leakage but fully exposing initialized private data under high IPC.** The distillation process involves matching aggregated information from the original dataset, introducing randomness via iterative optimization with small batches of real data. In dataset distillation, the randomness introduced by the matching optimization is inherently applied to the initialized training samples, thereby protecting individual data information, particularly the private data used for initialization. We start by stating the objective function for matching:

$$\arg \min {}_\mathcal{S} \, \mathbb{E}_{\boldsymbol{\theta}_0 \sim \mathbf{P}_{\boldsymbol{\theta}}} \left[ \sum_{t=0}^{T-1} D(\xi(\mathcal{S}, \boldsymbol{\theta}^t), \xi(\mathcal{T}, \boldsymbol{\theta}^t)) \right] \quad \text{s.t.} \quad \boldsymbol{\theta}^{t+1} \leftarrow \boldsymbol{\theta}^t - \eta \cdot \nabla_{\boldsymbol{\theta}} \mathcal{L}_\mathcal{S}(\boldsymbol{\theta}^t) \,. \tag{4}$$

Here, the function $\xi(\cdot)$ maps datasets $\mathcal{S}$ or $\mathcal{T}$ into a common space, such as gradients, features, or trajectories. The distance function $D(\cdot, \cdot)$ measures the difference between these mappings.

To analyze how this optimization process contributes to differential privacy, we focus on the Distribution Matching (DM) approach (Zhao & Bilen, 2023), guided by recent advancements in privacy analysis (Dong et al., 2022; Carlini et al., 2022b). In there analysis, Dong et al. (2022) employ a linear feature extractor $\boldsymbol{\psi}_{\boldsymbol{\theta}} : \mathbb{R}^d \to \mathbb{R}^k$, defined as $\boldsymbol{\psi}_{\boldsymbol{\theta}}(\mathbf{x}) = \boldsymbol{\theta}\mathbf{x}$ for an input $\mathbf{x}$, where $\boldsymbol{\theta} \in \mathbb{R}^{k \times d}$. This extractor transforms inputs from both the distilled and original datasets into feature space, enabling the DM approach to match their distributions. This approach reveals the relationship between the finnal distilled dataset $\mathcal{S}^*$ and the original dataset $\mathcal{T}$, as shown in the following lemma:

**Lemma 1 (Connection between $\mathcal{S}^*$ and $\mathcal{T}$ (Dong et al., 2022))** . *For a real data initialization, if the optimized distilled dataset $\mathcal{S}^*$ is derived from $\mathcal{S} = \mathbf{s}_1, \cdots, \mathbf{s}_{|\mathcal{S}|}$ through distribution matching, then:*

$$\mathbf{s}_i^* = \mathbf{s}_i + \frac{1}{|\mathcal{T}|} \sum_{j=1}^{|\mathcal{T}|} \mathbf{x}_j - \frac{1}{|\mathcal{S}|} \sum_{j=1}^{|\mathcal{S}|} \mathbf{s}_j \in span(\mathcal{T}), \tag{5}$$

*where $span(\mathcal{T}) := \{\sum_{i=1}^{|\mathcal{T}|} w_i \mathbf{x}_i | 1 \leq i \leq |\mathcal{T}|, w_i \in \mathbb{R}, \mathbf{x}_i \in \mathcal{T}\}$ denotes the linear span of the dataset $\mathcal{T}$.*

**Remark 1** . *This lemma demonstrates that the distilled dataset $\mathcal{S}^*$, when derived through optimized matching, closely aligns with the distribution of $\mathcal{T}$. The proximity of $\mathcal{S}^*$ to $\mathcal{T}$ implies that as the size of $\mathcal{S}$ approaches that of $\mathcal{T}$, the distilled samples $\mathbf{s}_i^*$ resemble the original samples $\mathbf{s}_i$, thereby potentially increasing explicit privacy risks, as shown in Figure 2 (a).*

*The distilled dataset, derived through optimized matching from the initial data, can be conceptualized as a normal distribution with $\mu = \mathbf{s}_i + 1/|\mathcal{T}| \sum_{j=1}^{|\mathcal{T}|} \mathbf{x}_j - 1/|\mathcal{S}| \sum_{j=1}^{|\mathcal{S}|} \mathbf{s}_j$. Consequently, by comparing the Kullback-Leibler divergence between adjacent datasets, we can ascertain the privacy protection capabilities of the distilled dataset.*

Building upon Lemma 1, we utilize the concept of adjacent datasets from differential privacy to compare distributional differences. Our analysis reveals that dataset distillation inherently possesses property similar to differential privacy, as formalized in the following theorem (see our proof details in Appendix C):

**Theorem 1** . *Consider a target dataset $\mathcal{T}$ and a leave-one-out adjacent dataset $\mathcal{T}' = \mathcal{T} \setminus \{\mathbf{x}\}$, where $\mathbf{x}$ is not sampled for initialization in phase 1. The distilled datasets $\mathcal{S}$ and $\mathcal{S}'$, with $|\mathcal{S}| = |\mathcal{S}'| \ll |\mathcal{T}|$, show that the membership privacy leakage from removing $\mathbf{x}$ is bounded by:*

$$D_{KL}(P \parallel Q) \leq \frac{2B|\mathcal{S}|}{|\mathcal{T}|} \cdot \lambda_{\max}(\mathbf{\Sigma}^{-1}), \tag{6}$$

*where $P$ and $Q$ are the sample distributions of the distilled datasets $\mathcal{S}$ and $\mathcal{S}'$, respectively, $B$ is the upper bound value of the original data and $\lambda_{\max}$ is the largest eigenvalue of the inverse covariance matrix $\mathbf{\Sigma}$.*

Theorem 1 states that the differential privacy leakage introduced by the matching optimization is limited. However, it is important to note that while the matching process itself offers some privacy protection, the initialization phase can still pose initial data privacy risks. Notably, the majority of state-of-the-art distillation methods (Cazenavette et al., 2022; Guo et al., 2024; Sun et al., 2024) employ initialization with real data to improve performance, which leads to a significant privacy concern.

### 3.3 METHOD FOR EXPLICIT PRIVACY PROTECTION

As previously discussed, although dataset distillation can theoretically limit the leakage of individual data, initializing training samples can significantly expose privacy risks, especially under high IPC conditions, leading to explicit privacy leakage. To address this issue, we propose a plug-and-play module, termed Kaleidoscopic Transformation (KT), which introduce strong transformations to the selected real data during initialization. This module builds upon Differentiable Siamese Augmentation (DSA) (Zhao & Bilen, 2021), a promising approach originally designed to improve the generalization capabilities of distilled datasets. In our study, we adapt DSA as a transformation technique applied to the initialized real private

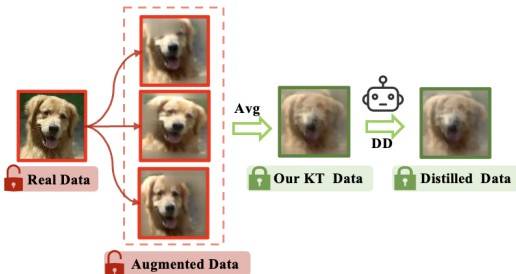

Figure 3: **Overview of Kaleidoscopic Transformation (KT).** We generate multiple augmented samples for each single input and then average them to obtain the final strongly augmented sample.

data. The randomness introduced by these transformations enhances the differential privacy property of the distilled dataset and provides better explicit privacy protection.

**Kaleidoscopic transformation.**  Consider the set $\mathcal{A}$ of all differentiable augmentations. Assume we have a sequence of image transformations $\{T_1, \ldots, T_i, \ldots, T_m\} \subset \mathcal{A}$, such as rotation, with each transformation $T_i$ associated with a probability $p_i$ of being executed. By leveraging these augmentations, we can generate a newly augmented dataset. The $j$-th augmented sample of the $i$-th example is:

$$\mathbf{s}'_{i,j} = \left( \circ_{k=1}^{m} T_k^{U_{i,j,k} \leq p_k} \right) (\mathbf{s}_i), \tag{7}$$

where for each transformation $T_i$, we generate a random variable $U_i \sim \text{Uniform}(0, 1)$. If $U_i \leq p_i$, $T_i$ is applied to the input image.

To enhance the transformation process, we produce $n$ augmented samples for each input and derive the final augmented sample by averaging: $\mathbf{s}'_i = \frac{1}{n} \sum_{j=1}^{n} \mathbf{s}'_{i,j}$. As illustrated in Figure 3 , employing multiple data augmentations can substantially improve privacy protection. Therefore, we initialize the distilled dataset using transformed samples $\mathbf{s}'$, rather than the original samples $\mathbf{s}$.

Note that KT not only enhances explicit privacy of the distilled dataset but also introduces additional randomness into the distillation process, thereby strengthening the differential privacy property of the resulting dataset. We justify this by modeling a differential transformation as a random bounded perturbation $\boldsymbol{\epsilon}$ (Rajput et al., 2019), with $\|\boldsymbol{\epsilon}\| \leq \epsilon_0$ and $\|T(\mathbf{s}) - \mathbf{s}\| \leq \epsilon_0$. It allows modeling the distribution of the distilled dataset obtained through KT, therefore enabling calculating the KL divergence between adjacent datasets. The comparison of differential privacy property of KT with those of the original distillation process is demonstrated in Theorem 2 (see proof details in Appendix D ):

---

**Theorem 2 .** *Consider a target dataset $\mathcal{T}$ and a leave-one-out dataset $\mathcal{T}' = \mathcal{T} \setminus \mathbf{x}$, where $\mathbf{x}$ is not used for initialization in phase 1. The* KT *initialized distilled datasets $\mathcal{S}_{\text{KT}}$ and $\mathcal{S}'_{\text{KT}}$, with $|\mathcal{S}_{\text{KT}}| = |\mathcal{S}'_{\text{KT}}| \ll |\mathcal{T}|$, show that the membership privacy leakage from removing $\mathbf{x}$ is bounded by:*

$$\mathrm{D}_{\mathrm{KL}}(P_{\mathrm{KT}} \parallel Q_{\mathrm{KT}}) \leq \frac{2B|\mathcal{S}|}{|\mathcal{T}|} \cdot \lambda_{\max}((\mathbf{\Sigma} + \nicefrac{1}{n}\mathbf{\Sigma}_{\boldsymbol{\epsilon}})^{-1}) < \mathrm{D}_{\mathrm{KL}}(P \parallel Q), \tag{8}$$

*where $P_{\mathrm{KT}}$ and $Q_{\mathrm{KT}}$ are the sample distributions of the distilled datasets $\mathcal{S}_{\mathrm{KT}}$ and $\mathcal{S}'_{\mathrm{KT}}$.*

---

We further demonstrate in Proposition 2 that though KT introduces perturbations to samples during the dataset distillation initialization phase, it maintains the similar efficacy as real data initialization.

---

**Proposition 2 .** *For a sample $\mathbf{s}_i$ randomly selected from the real dataset, the bound for the transformed data $\mathbf{s}'_i$ is:*

$$\|\text{KT}(\mathbf{s}_i) - \mathbf{s}_i\| = \|\mathbf{s}'_i - \mathbf{s}_i\| = \frac{1}{n} \left\| \sum_{j=1}^{n} \mathbf{s}'_{i,j} - \mathbf{s}_i \right\| \leq \frac{1}{n} \sum_{j=1}^{n} \left\| \mathbf{s}'_{i,j} - \mathbf{s}_i \right\| \leq \epsilon_0. \tag{9}$$

---

Therefore, the proposed KT not only enhances explicit privacy and differential privacy property, but also preserves the effectiveness comparable to real data initialization.

## 4 EXPERIMENT

### 4.1 EXPERIMENT SETUP

**Datasets and Neural Networks:**  We conduct experiments on both small-scale and large-scale datasets. For small-scale data, we evaluate our method on CIFAR-10 ($32 \times 32$) (Krizhevsky et al., 2009b) and CIFAR-100 ($32 \times 32$) (Krizhevsky et al., 2009a). For large-scale data, we conduct experiments on Tiny-ImageNet ($64 \times 64$) (Le & Yang, 2015), to assess the scalability and effectiveness of our method on more complex and varied datasets.

Following previous dataset distillation studies (Yin et al., 2023; Sun et al., 2024; Guo et al., 2024), we employ ConvNet (Guo et al., 2024) as our backbone architectures across all datasets. For ConvNet, specifically, Conv-3 is employed for CIFAR-10/100, while Conv-4 is used for Tiny-ImageNet.

Table 1: Comparison with previous dataset distillation methods on CIFAR-100 and Tiny ImageNet. **Membership Privacy** and **Explicit Privacy** are evaluated via TPR@0.1% FPR and LPIPS, respectively.

| | Method | TPR@0.1%FPR (↓) | | | Average LPIPS Distance (↑) | | | Test Accuracy (↑) | | |
|---|---|---|---|---|---|---|---|---|---|---|
| | | 1 | 10 | 50 | 1 | 10 | 50 | 1 | 10 | 50 |
| **CIFAR-100** | Full Dataset | | $24.8 \pm 0.4^*$ | | | $0^*$ | | | $61.27^*$ | |
| | DM | $0.11 \pm 0.02$ | $0.18 \pm 0.01$ | $0.9 \pm 0.1$ | 0.41 | 0.30 | 0.24 | $11.4 \pm 0.3$ | $29.7 \pm 0.3$ | $43.6 \pm 0.4$ |
| | KT-DM | $0.11 \pm 0.01$ | $0.16 \pm 0.02$ | $0.42 \pm 0.05$ | 0.43 | 0.35 | 0.33 | $7.8 \pm 0.1$ | $24.1 \pm 0.2$ | $40.2 \pm 0.3$ |
| | DSA | $0.11 \pm 0.02$ | $0.19 \pm 0.01$ | $1.3 \pm 0.1$ | 0.41 | 0.27 | 0.19 | $13.9 \pm 0.4$ | $32.4 \pm 0.3$ | $38.6 \pm 0.3$ |
| | KT-DSA | $0.1 \pm 0.03$ | $0.17 \pm 0.02$ | $0.45 \pm 0.03$ | 0.44 | 0.34 | 0.36 | $8.2 \pm 0.3$ | $26.5 \pm 0.2$ | $35.3 \pm 0.2$ |
| | MTT | $0.1 \pm 0.02$ | $0.19 \pm 0.05$ | $1.8 \pm 0.1$ | 0.38 | 0.24 | 0.09 | $24.3 \pm 0.3$ | $39.7 \pm 0.4$ | $47.7 \pm 0.2$ |
| | KT-MTT | $0.1 \pm 0.02$ | $0.16 \pm 0.02$ | $0.5 \pm 0.2$ | 0.39 | 0.35 | 0.33 | $22.1 \pm 0.2$ | $34.6 \pm 0.3$ | $42.8 \pm 0.3$ |
| | DATM | $0.13 \pm 0.03$ | $0.4 \pm 0.05$ | $\mathbf{3.2 \pm 0.1}$ | 0.36 | 0.20 | $\mathbf{0.02}$ | $27.9 \pm 0.2$ | $47.2 \pm 0.4$ | $\mathbf{55.0 \pm 0.2}$ |
| | KT-DATM | $0.1 \pm 0.02$ | $0.16 \pm 0.02$ | $\mathbf{0.6 \pm 0.2}$ | 0.37 | 0.34 | $\mathbf{0.31}$ | $22.8 \pm 0.2$ | $40.2 \pm 0.3$ | $\mathbf{49.2 \pm 0.3}$ |
| | RDED | $0.14 \pm 0.02$ | $0.44 \pm 0.05$ | $\mathbf{3.4 \pm 0.1}$ | 0.04 | 0.02 | $\mathbf{0.01}$ | $19.6 \pm 0.3$ | $48.1 \pm 0.3$ | $\mathbf{57.0 \pm 0.1}$ |
| | KT-RDED | $0.1 \pm 0.02$ | $0.17 \pm 0.01$ | $\mathbf{0.6 \pm 0.06}$ | 0.28 | 0.28 | $\mathbf{0.27}$ | $13.2 \pm 0.4$ | $40.2 \pm 0.3$ | $\mathbf{54.1 \pm 0.5}$ |
| **Tiny-ImageNet** | Full Dataset | | $17.3 \pm 0.5^*$ | | | $0^*$ | | | $49.73^*$ | |
| | DM | $0.1 \pm 0.02$ | $0.15 \pm 0.05$ | $0.9 \pm 0.2$ | 0.43 | 0.33 | 0.19 | $3.9 \pm 0.2$ | $12.9 \pm 0.4$ | $24.1 \pm 0.3$ |
| | KT-DM | $0.1 \pm 0.02$ | $0.15 \pm 0.02$ | $0.3 \pm 0.04$ | 0.43 | 0.39 | 0.35 | $2.2 \pm 0.2$ | $9.1 \pm 0.2$ | $22.7 \pm 0.3$ |
| | DSA | — | — | — | — | — | — | — | — | — |
| | KT-DSA | — | — | — | — | — | — | — | — | — |
| | MTT | $0.1 \pm 0.02$ | $0.17 \pm 0.04$ | $1.1 \pm 0.2$ | 0.41 | 0.23 | 0.05 | $8.8 \pm 0.3$ | $23.2 \pm 0.2$ | $28.0 \pm 0.3$ |
| | KT-MTT | $0.1 \pm 0.02$ | $0.16 \pm 0.02$ | $0.5 \pm 0.2$ | 0.38 | 0.32 | 0.29 | $7.8 \pm 0.2$ | $20.4 \pm 0.1$ | $24.7 \pm 0.2$ |
| | DATM | $0.12 \pm 0.08$ | $0.2 \pm 0.04$ | $\mathbf{2.4 \pm 0.1}$ | 0.39 | 0.13 | $\mathbf{0.01}$ | $17.1 \pm 0.3$ | $31.1 \pm 0.3$ | $\mathbf{38.6 \pm 0.3}$ |
| | KT-DATM | $0.1 \pm 0.02$ | $0.16 \pm 0.02$ | $\mathbf{0.5 \pm 0.2}$ | 0.34 | 0.29 | $\mathbf{0.25}$ | $13.3 \pm 0.2$ | $27.6 \pm 0.3$ | $\mathbf{35.2 \pm 0.3}$ |
| | RDED | $0.12 \pm 0.04$ | $0.23 \pm 0.02$ | $\mathbf{2.8 \pm 0.1}$ | 0.04 | 0.03 | $\mathbf{0.01}$ | $12.0 \pm 0.1$ | $39.6 \pm 0.1$ | $\mathbf{49.6 \pm 0.2}$ |
| | KT-RDED | $0.11 \pm 0.01$ | $0.18 \pm 0.02$ | $\mathbf{0.6 \pm 0.07}$ | 0.22 | 0.23 | $\mathbf{0.20}$ | $7.6 \pm 0.3$ | $33.5 \pm 0.2$ | $\mathbf{47.3 \pm 0.2}$ |

**Baselines:** We evaluate our proposed method, KT, against a range of state-of-the-art techniques in both dataset distillation and data generator. For all experiments, we utilize three different random seeds and report both the mean and variance of the results.

- Dataset Distillation Methods: (1) distribution matching-based methods, such as DM (Zhao & Bilen, 2023); (2) gradient matching-based approaches, exemplified by DSA (Zhao & Bilen, 2021); (3) trajectory matching-based strategies, including MTT (Cazenavette et al., 2022) and DATM (Guo et al., 2024); and (4) non-optimization-based frameworks like RDED (Sun et al., 2024).
- Data Generator Methods: (1) DP GAN-based methods, such as DP-MEPF (Harder et al., 2022); (2) DP distillation-based methods, such as PSG (Chen et al., 2022).

**MIA Settings and Attack Metrics.** We consider a typical scenario where the adversary possesses access to the distilled dataset $\mathcal{S}$ and employs it to train a target model $f_{\mathcal{S}}$. The objective of adversary is to infer membership information of the original dataset $\mathcal{T}$.

For our membership inference attack framework on distilled datasets, we address a critical oversight in previous works (Dong et al., 2022; Carlini et al., 2022b) that incorrectly treated training data not used for initialization as non-members. We consider the entire original training set as members of the distilled dataset, as all samples contribute to the distillation process. To ensure fairness, we employ identical test samples and shadow models across various distilled and original datasets (see Figure 7 in Appendix E.3 for a detailed illustration of our framework). Following Carlini et al. (2022a), we use TPR @ 0.1% FPR as the success criterion for membership inference attacks.

Further comprehensive experimental configurations, including detailed settings aligned with the original distillation methods and specific hyperparameter choices, are provided in Appendix E.

## 4.2 DIFFERENTIAL PRIVACY-LIKE PROPERTIES OF DISTILLED DATASETS AGAINST MEMBERSHIP INFERENCE

**Comparison with State-of-the-Art Dataset Distillation Methods.** We use TPR@0.1% FPR (Carlini et al., 2022a) to evaluate the differential privacy of distilled datasets, focusing on attack success at low false positive rates. It is evident that LiRA successfully attacks all three full datasets, as shown in Table 1. However, models trained on distilled datasets, even without employing the our KT method, substantially reduces the attack success rate. *The results confirms that distilled datasets can ensure differential privacy, aligning with our analysis in Section 3.2.* Notably, when KT is applied, the attack success rate continues to decrease, further verifying that KT enhances differential privacy. Detailed results for CIFAR-10 can be found in Appendix Appendix F.

Table 2: Comparison with previous **data generation methods** on CIFAR-10.

| | Method | TPR@0.1%FPR (↓) | | | Average LPIPS Distance (↑) | | | Test Accuracy (↑) | | |
|---|---|---|---|---|---|---|---|---|---|---|
| | | 1 | 10 | 50 | 1 | 10 | 50 | 1 | 10 | 50 |
| CIFAR-10 | DP-MEPF($\epsilon = 10$) | $0.1 \pm 0.01$ | $0.13 \pm 0.01$ | $0.16 \pm 0.02$ | 0.40 | 0.38 | 0.35 | $16.6 \pm 0.4$ | $24.1 \pm 0.3$ | $28.0 \pm 0.2$ |
| | PSG($\epsilon = 10$) | $0.1 \pm 0.02$ | $0.12 \pm 0.03$ | $0.15 \pm 0.02$ | 0.42 | 0.38 | 0.34 | $28.9 \pm 0.4$ | $40.3 \pm 0.5$ | $47.2 \pm 0.2$ |
| | KT-DATM | $0.1 \pm 0.02$ | $0.14 \pm 0.02$ | $0.4 \pm 0.1$ | 0.36 | 0.35 | 0.33 | $43.3 \pm 0.2$ | $62.3 \pm 0.1$ | $\mathbf{69.2 \pm 0.2}$ |
| | KT-RDED | $0.12 \pm 0.01$ | $0.18 \pm 0.03$ | $0.7 \pm 0.1$ | 0.35 | 0.34 | 0.31 | $17.7 \pm 0.2$ | $42.2 \pm 0.2$ | $\mathbf{62.5 \pm 0.3}$ |

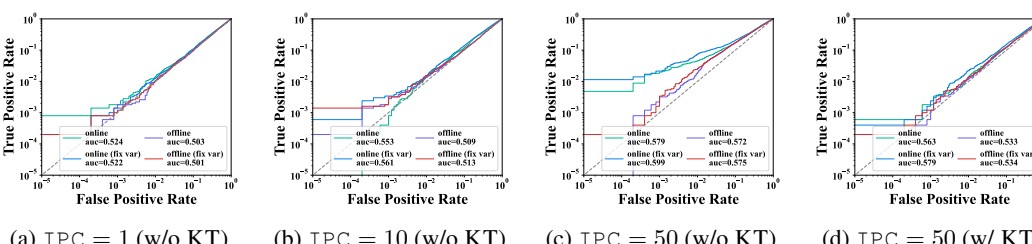

(a) IPC = 1 (w/o KT)  (b) IPC = 10 (w/o KT)  (c) IPC = 50 (w/o KT)  (d) IPC = 50 (w/ KT)

Figure 4: ROC curve graphs of DATM on TinyImageNet at different IPC values: With higher IPC, the success rate of attacks at low false positive rates increases. The application of KT at IPC = 50 demonstrates a significant reduction in attack success rate.

**Comparison with State-of-the-Art Data Generator Methods.** We further compare our method KT with existing data generation techniques designed for differential privacy and explicit privacy, as illustrated in Table 2 . Our experiments focus on CIFAR-10, as it is the primary benchmark for most DP data generation methods. Other datasets like Tiny-ImageNet are often treated as public data by some methods (Wang et al., 2024; Lin et al., 2023), precluding a fair comparison.

*Our approach demonstrates a balanced performance in privacy preservation and data utility.* While methods like PSG and DP-MEPF exhibit strong privacy guarantees due to their strict privacy budgets and noise initialization, they struggle with data utility, particularly in downstream tasks requiring model training from scratch under the same IPC. We conducted experiments on the baseline of the DP-generator for more $\epsilon$ values and plotted the trade-off curves in Appendix G , demonstrating that KT-DATM offers better data availability under comparable MIA defense.

It is important to note that dataset distillation inherently aims to generate smaller, more efficient datasets. The privacy protection it brings is an additional benefit. *Additionally, the computational overhead introduced by the KT plugin is negligible compared to that of the DP generator.*

**Impact of Varying IPC on Resisting MIA.** We perform experiments on the Tiny-ImageNet dataset, utilizing DATM (Guo et al., 2024) to obtain distilled datasets with IPC values of 1, 10, and 50. Subsequently, we apply LiRA membership inference attacks, with results illustrated in Figure 4 . As the IPC value increases, AUC of LiRA's ROC curves show also increase, which suggests that higher IPC values reduce the differential privacy protection of the distilled datasets. Furthermore, for a high IPC of 50, we compare scenarios with and without our KT. The results presented in Figure 4 (c) and (d), show that our KT reduces the AUC scores of the ROC curves, demonstrating that *our* KT *effectively enhances differential privacy, even at elevated* IPC *levels.*

**Membership Privacy of Initialization.** We are concerned about the privacy leakage of the training samples used for initialization. In Appendix H , we experimented with the fix-target membership inference attack (Ye et al., 2022). The KT plugin not only protects the explicit privacy of the initialization samples but also defends against MIA.

### 4.3 ENHANCED EXPLICIT PRIVACY UNDER HIGH IPC VIA KT

**Comparison with State-of-the-Art Methods.** We utilize the Learned Perceptual Image Patch Similarity (LPIPS) metric (Zhang et al., 2018) to estimate explicit privacy leakage. For a distilled dataset, we compute the average LPIPS distance from its corresponding real sample set to quantify privacy leakage. A larger LPIPS distance signifies enhanced explicit privacy protection.

As demonstrated in Table 1 , as the IPC increases, LPIPS significantly decrease. *This suggests that higher* IPC *more severely exposure explicit privacy, consistent with our analysis in Section 3.2 .*

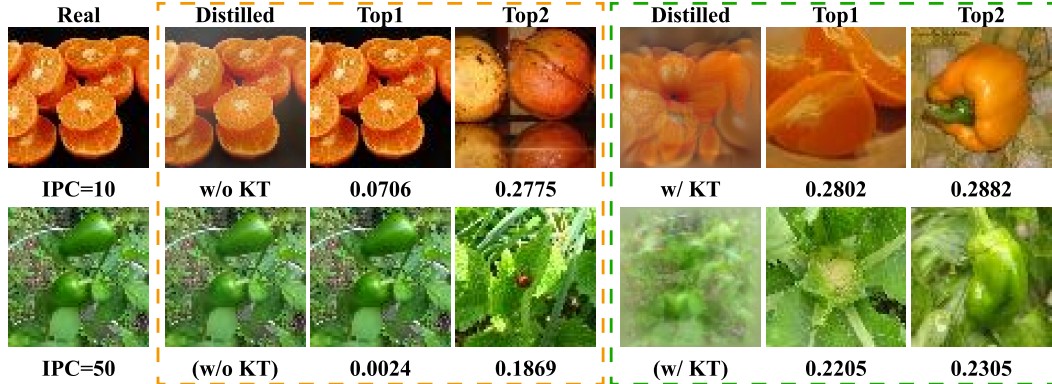

Figure 5: **DATM presents explicit privacy protection at IPC=10 and 50.** The orange and green regions represent the explicit privacy measurement results of the distilled samples without and with the KT plugin, respectively. We selected the top 2 most similar original data points, with the values measured using LPIPS.

Furthermore, we visualize samples of the distilled dataset and identify the top-2 nearest samples from the original dataset in Figure 5. At `IPC` = 10 and 50, the distilled dataset without our method completely leaks the private data used for initialization, indicating significant explicit privacy leakage. *However, with the introduction of* KT*, the distilled samples are visually distinct from their nearest neighbors in the original dataset, demonstrating enhanced explicit privacy.*

**Influence of Hyper-parameter** $n$. To determine the optimal setting for the KT hyper-parameter $n$, we conducted experiments varying $n$ from 1 to 5 with KT-DATM on TinyImageNet using `IPC = 50`. *Our findings reveal a critical trade-off between privacy protection and data utility.* At $n = 1$, KT behaves like data augmentation, offering insufficient privacy protection. For $n \geq 4$, privacy improves but data utility sharply declines. Empirically, we found $n = 3$ to be the optimal balance between enhancing privacy and maintaining utility.

## 5 CONCLUSION AND LIMITATION

**Conclusion.** In this study, we first identify that the distilled datasets produced by state-of-the-art distillation methods strongly resemble to real data, indicating significant privacy leakage, termed as explicit privacy leakage. We further provide a theoretical analysis showing that while distilled datasets can achieve differential privacy, a high `IPC` can undermine both differential privacy and explicit privacy. We identify that the primary source of privacy leakage in distilled data is traced to the initialization of distilled images using real data. Building on these insights, we propose a plug-and-play module, Kaleidoscopic Transformation (KT), which introduces enhanced perturbations to the selected real data

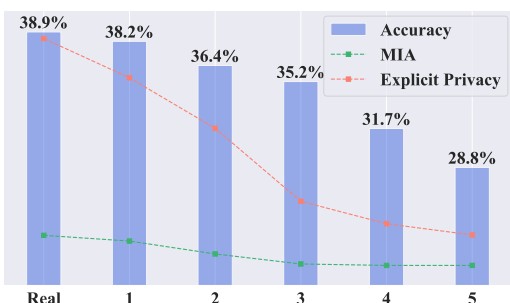

Figure 6: **Impact of KT Parameter $n$ on Privacy and Utility.** The graph illustrates how varying $n$ from 1 to 5 affects explicit privacy protection and data utility, revealing an optimal trade-off at $n = 3$.

during the initialization phase. Extensive experiments have verfied that our method KT is able to ensure both differential privacy and explicit privacy, while preserving the generalization performance of the distilled data.

**Limitation.** The effectiveness of KT in downstream task accuracy is constrained by the underlying dataset distillation algorithm. While KT can be integrated as a plugin into existing dataset distillation methods to provide cost-free privacy protection, it does not improve the distillation quality for model training from scratch. Our experiments show that RDED-KT outperforms DATM-KT in downstream accuracy, reflecting the base algorithm's capability in preserving task-relevant information. Thus, KT's impact on model performance is inherently tied to the efficacy of the chosen distillation method.

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

# A    RELATED WORK

## A.1    DATASET DISTILLATION

Current solutions can be categorized based on their optimization mechanisms (Lei & Tao, 2023): (1) *Meta-Learning Framework*: Distilled data are considered as hyperparameters, which are optimized in a nested loop according to the distilled-data-trained model's risk with respect to (*w.r.t.*) the original data, including DD (Wang et al., 2018), KIP (Nguyen et al., 2021) and FRePo (Zhou et al., 2022). (2) *Gradient Matching*: Aims to match the network gradients computed by the original dataset and the distilled dataset, including DC (Zhao et al., 2020), DSA (Zhao & Bilen, 2021), and IDC (Kim et al., 2022). (3) *Distribution Matching*: Directly matches the distribution of original dataset and distilled data. Methods in this category includ DM (Zhao & Bilen, 2023), CAFE (Wang et al., 2022a), SRe$^2$L (Yin et al., 2023). (4) *Trajectory Matching*: Matches the training trajectories of models trained on original and distilled data over multiple steps. This category includes MTT (Cazenavette et al., 2022) and , DATM (Guo et al., 2024). The above methods are based on optimization. Notably, RDED (Sun et al., 2024) introduces an optimization-free paradigm, which directly crop and select realistic patches from the original data, and then stitch the selected patches into the new images as the distilled dataset. It achieves remarkable performance, particularly with large-scale and high-resolution datasets.

## A.2    LIRA

Specifically, the privacy attack LiRA encompasses three stages. Firstly, the adversary randomly samples $N$ datasets from natural distribution to train shadow models. Therefore, for each data sample, there are $N/2$ shadow models trained on it (*the IN models*) and another $N/2$ that are not trained on it (*the OUT models*). Secondly, the adversary estimates the means $\boldsymbol{\mu}_{\text{in}}, \boldsymbol{\mu}_{\text{out}}$, and the variances $\boldsymbol{\sigma}_{\text{in}}^2, \boldsymbol{\sigma}_{\text{out}}^2$ of model confidence for the IN and OUT models, respectively. Finally, to attack, the adversary queries the victim model $f$ with a target example $(\mathbf{x}, y)$ to estimate the likelihood $\Lambda$, defined as:

$$\Lambda := \frac{p(\text{conf}_{\text{obs}} \mid \mathcal{N}(\boldsymbol{\mu}_{\text{in}}, \boldsymbol{\sigma}_{\text{in}}^2))}{p(\text{conf}_{\text{obs}} \mid \mathcal{N}(\boldsymbol{\mu}_{\text{out}}, \boldsymbol{\sigma}_{\text{out}}^2))} , \tag{10}$$

where $\text{conf}_{\text{obs}} = \log\left(f(\mathbf{x})_y / 1 - f(\mathbf{x})_y\right)$ is the confidence of target model $f$ on the test example $(\mathbf{x}, y)$. Here, $f(\mathbf{x})_y$ represents the probability assigned by the target model $f$ to the true membership label $y$ when evaluating the attack test example $\mathbf{x}$.

Note that LiRA determines if a data point was part of the training set by comparing a calculated likelihood score $\Lambda$ to a predetermined threshold $\tau$. If $\Lambda > \tau$, the data point is classified as a member of the training set.

# B    PROOF OF PROPOSITION 1

*Proof.* Suppose a full dataset $\mathcal{T}$ and an adjacent dataset $\mathcal{T}'$ which differ in one sample. Let $\mathcal{M}$ be the random sample mechanism that randomly returns a subset of the data without replacement. Let $\mathcal{S}_0, \mathcal{S}_1$ and $\mathcal{S}$ denote the all subsets in $\mathcal{M}(\mathcal{T}), \mathcal{M}(\mathcal{T}')$ and the joint domain of them respectively. For a random subset $S \in \mathcal{S}$, we have

$$\Pr(\mathcal{M}(\mathcal{T}) = S) = \begin{cases} \frac{1}{\binom{|\mathcal{T}|}{|\mathcal{S}|}}, & S \in \mathcal{S}_0, \\ 0, & \text{otherwise.} \end{cases} \tag{11}$$

$$\Pr(\mathcal{M}(\mathcal{T}') = S) = \begin{cases} \frac{1}{\binom{|\mathcal{T}'|}{|\mathcal{S}|}}, & S \in \mathcal{S}_1, \\ 0, & \text{otherwise.} \end{cases} \tag{12}$$

**case 1** ($|\mathcal{T}'| = |\mathcal{T}| + 1$) **:** Due to $\mathcal{T} \subset \mathcal{T}'$, then we have

$$\Pr(\mathcal{M}(\mathcal{T}) \in \mathcal{S}_0) = 1, \tag{13}$$

$$\Pr(\mathcal{M}(\mathcal{T}') \in \mathcal{S}_0) = \frac{\binom{|\mathcal{T}|}{|\mathcal{S}|}}{\binom{|\mathcal{T}'|}{|\mathcal{S}|}} = \frac{\binom{|\mathcal{T}|}{|\mathcal{S}|}}{\binom{|\mathcal{T}|+1}{|\mathcal{S}|}}. \tag{14}$$

We calculate this case based on the definition of differential privacy.

$$
\begin{aligned}
\Pr(\mathcal{M}(\mathcal{T}) \in \mathcal{S}) &= \Pr(\mathcal{M}(\mathcal{T}) \in \mathcal{S}_0) + \Pr(\mathcal{M}(\mathcal{T}) \in \mathcal{S}/\mathcal{S}_0) \\
&= \Pr(\mathcal{M}(\mathcal{T}) \in \mathcal{S}_0) + 0 \\
&= \Pr(\mathcal{M}(\mathcal{T}') \in \mathcal{S}_0) \cdot \frac{\binom{|\mathcal{T}|+1}{|\mathcal{S}|}}{\binom{|\mathcal{T}|}{|\mathcal{S}|}} \\
&= \Pr(\mathcal{M}(\mathcal{T}') \in \mathcal{S}_0) \cdot \frac{|\mathcal{T}|+1}{|\mathcal{T}|-|\mathcal{S}|+1} \\
&\le \Pr(\mathcal{M}(\mathcal{T}') \in \mathcal{S}) \cdot \frac{|\mathcal{T}|+1}{|\mathcal{T}|-|\mathcal{S}|+1}
\end{aligned}
\tag{15}
$$

**case 2** $(|\mathcal{T}'| = |\mathcal{T}| - 1)$ **:** Due to $\mathcal{T}' \subset \mathcal{T}$, then we have

$$
\Pr(\mathcal{M}(\mathcal{T}) \in \mathcal{S}_1) = \frac{\binom{|\mathcal{T}'|}{|\mathcal{S}|}}{\binom{|\mathcal{T}|}{|\mathcal{S}|}} = \frac{\binom{|\mathcal{T}|-1}{|\mathcal{S}|}}{\binom{|\mathcal{T}|}{|\mathcal{S}|}},
\tag{16}
$$

$$
\Pr(\mathcal{M}(\mathcal{T}') \in \mathcal{S}_1) = 1.
\tag{17}
$$

We calculate this case based on the definition of differential privacy.

$$
\begin{aligned}
\Pr(\mathcal{M}(\mathcal{T}) \in \mathcal{S}) &= \Pr(\mathcal{M}(\mathcal{T}) \in \mathcal{S}_1) + \Pr(\mathcal{M}(\mathcal{T}) \in \mathcal{S}/\mathcal{S}_1) \\
&= \Pr(\mathcal{M}(\mathcal{T}) \in \mathcal{S}_1) + \frac{|\mathcal{S}|}{|\mathcal{T}|} \\
&= \Pr(\mathcal{M}(\mathcal{T}') \in \mathcal{S}_1) \cdot \frac{\binom{|\mathcal{T}|-1}{|\mathcal{S}|}}{\binom{|\mathcal{T}|}{|\mathcal{S}|}} + \frac{|\mathcal{S}|}{|\mathcal{T}|} \\
&= \Pr(\mathcal{M}(\mathcal{T}') \in \mathcal{S}_1) \cdot \frac{|\mathcal{T}|-|\mathcal{S}|}{|\mathcal{T}|} + \frac{|\mathcal{S}|}{|\mathcal{T}|} \\
&\le \Pr(\mathcal{M}(\mathcal{T}') \in \mathcal{S}) \cdot \frac{|\mathcal{T}|-|\mathcal{S}|}{|\mathcal{T}|} + \frac{|\mathcal{S}|}{|\mathcal{T}|}
\end{aligned}
\tag{18}
$$

We combine case 1 and case 2, and we have $e^\epsilon = \max(\frac{|\mathcal{T}|+1}{|\mathcal{T}|-|\mathcal{S}|+1}, \frac{|\mathcal{T}|-|\mathcal{S}|}{|\mathcal{T}|}) = \frac{|\mathcal{T}|+1}{|\mathcal{T}|-|\mathcal{S}|+1}$, and $\delta = \max(0, \frac{|\mathcal{S}|}{|\mathcal{T}|}) = \frac{|\mathcal{S}|}{|\mathcal{T}|}$. Therefore, randomly sampling $|\mathcal{S}|$ samples from the original dataset (and using them to initialize the distilled dataset) satisfies $(\ln \frac{|\mathcal{T}|+1}{|\mathcal{T}|-|\mathcal{S}|+1}, \frac{|\mathcal{S}|}{|\mathcal{T}|})$-differential privacy. $\qquad\square$

## C  Proof of Theorem 1

*Proof.* The distribution of individual samples in the distilled dataset can be modeled as a normal distribution.

> **Assumption 1 .** *We assume the data of $\mathcal{T}$ and $\mathcal{S}$ are bounded, i.e.,*
>
> $$
> \exists B > 0, \forall \mathbf{x} \in \mathcal{T} \cup \mathcal{S}, \|\mathbf{x}\|_2 \le B.
> \tag{19}
> $$
>
> *For a particular sample $\mathcal{S}_i^*$ in the distilled dataset, to account for the matching stochasticity, we have*
>
> $$
> \mathbf{s}_i^* \sim \mathcal{N}\!\left(\mathbf{s}_i + \frac{1}{|\mathcal{T}|} \sum_{j=1}^{|\mathcal{T}|} \mathbf{x}_j - \frac{1}{|\mathcal{S}|} \sum_{j=1}^{|\mathcal{S}|} \mathbf{s}_j, \, \boldsymbol{\Sigma}_i\right).
> \tag{20}
> $$

Suppose a full dataset $\mathcal{T}$ and an adjacent dataset $\mathcal{T}'$ which differ in one sample $\mathbf{x}_{\text{differ}}$, such that $\mathbf{x}_{\text{differ}}$ is not used for initialization. The distilled dataset are $\mathcal{S}$ and $\mathcal{S}'$ and $|\mathcal{S}| = |\mathcal{S}'| \ll |\mathcal{T}|$. The distribution of sample $\mathbf{s}_i^*$ within the distilled dataset can be denoted as $p(\mathbf{s}_i^*) = \mathbb{P}(\mathbf{s}_i^* | \mathcal{T})$ and $q(\mathbf{s}_i^*) = \mathbb{P}(\mathbf{s}_i^* | \mathcal{T}')$.

Due to the difference in $\mathbf{x}_{\text{differ}}$, the privacy variations introduced during the matching process can be represented as KL divergence between the two distributions:

$$
\begin{aligned}
D_{KL}(p \parallel q) &= \frac{1}{2}\left(\text{tr}(\boldsymbol{\Sigma}_i^{-1}\boldsymbol{\Sigma}_i) + (\boldsymbol{\mu}_i' - \boldsymbol{\mu}_i)^T\boldsymbol{\Sigma}_i^{-1}(\boldsymbol{\mu}_i' - \boldsymbol{\mu}_i) - n - \log\frac{\det\boldsymbol{\Sigma}_i}{\det\boldsymbol{\Sigma}_i}\right) \\
&= \frac{1}{2}(\boldsymbol{\mu}_i' - \boldsymbol{\mu}_i)^T\boldsymbol{\Sigma}_i^{-1}(\boldsymbol{\mu}_i' - \boldsymbol{\mu}_i) \\
&\leq \|\boldsymbol{\mu}_i' - \boldsymbol{\mu}_i\|_2 \cdot \lambda_{\max}(\boldsymbol{\Sigma}_i^{-1}).
\end{aligned}
\tag{21}
$$

where $n$ is the dimension of $\mathbf{x}$, $\lambda_{\max}$ is the largest eigenvalue of the covariance matrix $\boldsymbol{\Sigma}$ and

$$
\begin{aligned}
\|\boldsymbol{\mu}_i' - \boldsymbol{\mu}_i\|_2 &= \left\|\frac{1}{|\mathcal{T}| - 1}\sum_{j=1}^{|\mathcal{T}|-1}\mathbf{x}_j - \frac{1}{|\mathcal{T}|}\sum_{j=1}^{|\mathcal{T}|}\mathbf{x}_j\right\|_2 \\
&= \frac{1}{|\mathcal{T}|}\left\|\frac{1}{|\mathcal{T}| - 1}\sum_{j=1}^{|\mathcal{T}|-1}\mathbf{x}_j - \mathbf{x}_{\text{differ}}\right\|_2.
\end{aligned}
\tag{22}
$$

According to Assumption 1, we have $\|\mathbf{x}\|_2 \leq B$ for all $\mathbf{x} \in \mathcal{T} \cup \mathcal{S}$. Therefore, we have

$$
\left\|\frac{1}{|\mathcal{T}| - 1}\sum_{j=1}^{|\mathcal{T}|-1}\mathbf{x}_j - \mathbf{x}_{\text{differ}}\right\|_2 \leq \left\|\frac{1}{|\mathcal{T}| - 1}\sum_{j=1}^{|\mathcal{T}|-1}\mathbf{x}_j\right\|_2 + \|\mathbf{x}_{\text{differ}}\|_2 \leq 2B.
\tag{23}
$$

From previous analysis, it can be concluded that the KL divergence of the distillation results from adjacent datasets is bounded:

$$
D_{KL}(p \parallel q) \leq \frac{2B}{|\mathcal{T}|} \cdot \lambda_{\max}(\boldsymbol{\Sigma}_i^{-1}).
\tag{24}
$$

The total KL divergence of the distilled dataset also can be bounded:

$$
D_{KL}(P \parallel Q) \leq \frac{2B|\mathcal{S}|}{|\mathcal{T}|} \cdot \lambda_{\max}(\boldsymbol{\Sigma}^{-1}).
\tag{25}
$$

where $P$ and $Q$ are the joint distributions of the adjacent datasets and $\lambda_{\max}(\boldsymbol{\Sigma}^{-1})$ corresponds to the largest eigenvalue of the covariance matrix across all samples in the distilled dataset. $\qquad\square$

# D    PROOF OF THEOREM 2

*Proof.* As demonstrated in the proof of Theorem 1, $\mathcal{T}$ and $\mathcal{T}'$ are adjacent datasets where $\mathcal{T}' = \mathcal{T} \setminus \mathbf{x}_{\text{differ}}$. In section 3.3, we establish the relationship between the KT-initialized distilled data $\mathbf{s}_i'$ and the initialized real data $\mathbf{s}_i$.

$$
\mathbf{s}_i' = \frac{1}{n}\sum_{j=1}^{n}\left(\circ_{k=1}^{m} T_k^{U_{i,j,k} \leq p_k}\right)(\mathbf{s}_i).
\tag{26}
$$

where $n$ is the We model the KT as a additive bounded noise $\bar{\boldsymbol{\epsilon}} = \sum_{j=1}^{n}\boldsymbol{\epsilon}_j$, where $\bar{\boldsymbol{\epsilon}} \sim \mathcal{N}(0, \frac{1}{n}\boldsymbol{\Sigma}_\epsilon)$, thus

$$
\mathbf{s}_i' = \mathbf{s}_i + \bar{\boldsymbol{\epsilon}}_i.
\tag{27}
$$

where $n$ represents the number of KT candidate transformation images, and $m$ represents the number of types of transformations. We can obtain the KT distilled dataset, optimized for matching as in Theorem 1, whose distribution can be represented as:

$$
\mathbf{s}_i'^* \sim \mathcal{N}\left(\mathbf{s}_i' + \bar{\boldsymbol{\epsilon}}_i + \frac{1}{|\mathcal{T}|}\sum_{j=1}^{|\mathcal{T}|}\mathbf{x}_j - \frac{1}{|\mathcal{S}|}\sum_{j=1}^{|\mathcal{S}|}(\mathbf{s}_j' + \bar{\boldsymbol{\epsilon}}_j), \boldsymbol{\Sigma}_i + \frac{1}{n}\boldsymbol{\Sigma}_\epsilon\right).
\tag{28}
$$

Recall the KL divergence upper bound, we have

$$D_{KL}(P_{\text{KT}} \parallel Q_{\text{KT}}) \leq \frac{2B\,|\mathcal{S}|}{|\mathcal{T}|} \cdot \lambda_{\max}((\boldsymbol{\Sigma} + \frac{1}{n}\boldsymbol{\Sigma}_\epsilon)^{-1}). \tag{29}$$

According to the matrix inversion lemma, for positive definite matrices:

$$\lambda_{\max}((\boldsymbol{\Sigma} + \frac{1}{n}\boldsymbol{\Sigma}\epsilon)^{-1}) < \lambda_{\max}(\boldsymbol{\Sigma}^{-1}). \tag{30}$$

Therefore, we have:

$$D_{KL}(P_{\text{KT}} \parallel Q_{\text{KT}}) < D_{KL}(P \parallel Q). \tag{31}$$

After KT initialization, the distillation difference caused by a single sample difference between adjacent datasets is smaller, thereby providing better differential privacy properties. □

## E  EXPRIMENTAL DETIALS

### E.1  IMPLEMENTATION DETAILS OF KT.

Our method use transformed data via KT instead of real samples for initialization. Notably, it does not involve modifying any distilling datasets process. Thus, our method is a plug-and-play approach that can be easily integrated into existing dataset distillation methods without requiring further modification. We utilize the source code[2] provided by the authors to obtain distilled data distill with `IPC` $\in \{1, 10, 50\}$.

### E.2  HYPERPARAMETER SETTINGS.

We provide detailed hyperparameter configurations for our distilled dataset evaluation in Figure 6. For Kaleidoscopic Transformation (KT), we empirically determined that setting $n = 3$ yields the optimal generalization performance, with probability thresholds for each transformation consistent with the DSA (Zhao & Bilen, 2021).

### E.3  A NEW MIA FRAMEWORK FOR DISTILLED DATASETS

Our membership inference attack framework for distilled datasets addresses the limitations of previous approaches by treating the entire original dataset as potential members. Figure 7 illustrates our unified evaluation method using LiRA, which employs common test samples for training shadow models.

This framework ensures a fair comparison across different distillation methods by using identical test samples and shadow models.

Our framework consists of three main steps:

- **Target Model Training:** We train the target model using the distilled dataset, following the same training procedure across all methods. We utilize the original dataset's training samples, designated as members, while the test set comprises non-members.

- **Shadow Model Training:** We train multiple shadow models, ensuring that each sample is treated as a member for half of the shadow models and as a non-member for the other half. To mitigate the potential impact of data augmentation on privacy, we apply DSA with multiple queries during this phase.

- **Attack Evaluation:** We input test cases into both the target and shadow models, computing scores to determine the attack results.

---

[2]DM and DSA: https://github.com/VICO-UoE/DatasetCondensation
MTT: https://github.com/GeorgeCazenavette/mtt-distillation
DATM: https://github.com/NUS-HPC-AI-Lab/DATM
RDED: https://github.com/LINs-lab/RDED

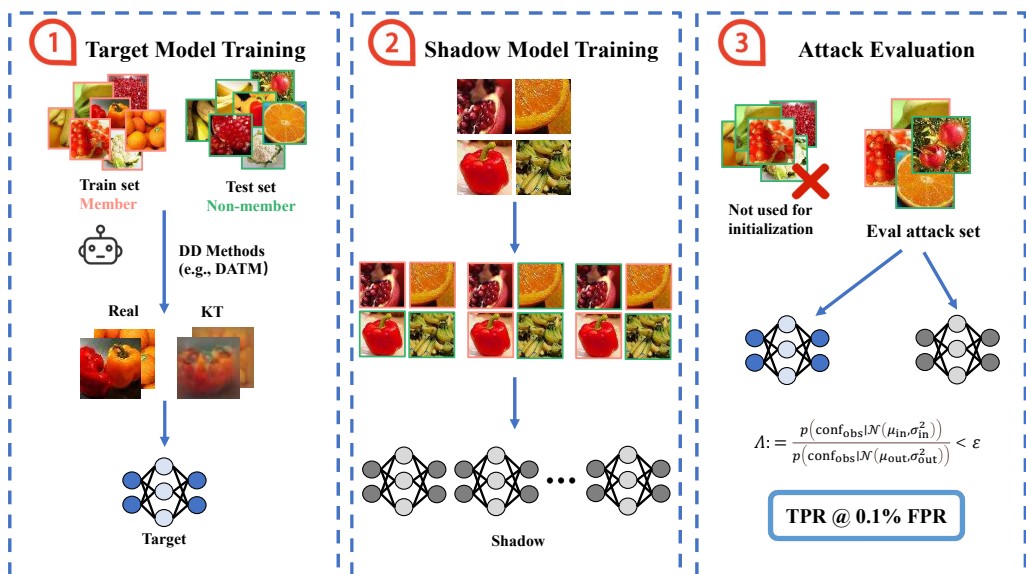

Figure 7: **Unified evaluation method of membership privacy using LiRA:** training shadow models using common test samples.

## F CIFAR-10 RESULTS IN 4.2

Table 3 presents a comprehensive comparison of our method with previous dataset distillation approaches on the CIFAR-10 dataset. We evaluate performance across three key metrics: membership privacy (measured by TPR@0.1% FPR), explicit privacy (measured by Average LPIPS Distance), and dataset utility (measured by Test Accuracy).

Table 3: Comparison with previous dataset distillation methods on CIFAR-10. **membership privacy** and **explicit privacy** are evaluated via TPR@0.1% FPR and LPIPS, respectively.

|  | Method | TPR@0.1%FPR ($\downarrow$) | | | Average LPIPS Distance ($\uparrow$) | | | Test Accuracy ($\uparrow$) | | |
|---|---|---|---|---|---|---|---|---|---|---|
|  |  | 1 | 10 | 50 | 1 | 10 | 50 | 1 | 10 | 50 |
| CIFAR-10 | Full Dataset | | $8.4 \pm 0.1^*$ | | | $0^*$ | | | $82.24^*$ | |
|  | DM | $0.08 \pm 0.02$ | $0.1 \pm 0.02$ | $0.6 \pm 0.05$ | $0.40$ | $0.36$ | $0.19$ | $26.0 \pm 0.8$ | $48.9 \pm 0.6$ | $63.0 \pm 0.4$ |
|  | KT-DM | $0.08 \pm 0.02$ | $0.1 \pm 0.03$ | $0.3 \pm 0.03$ | $0.41$ | $0.38$ | $0.36$ | $21.1 \pm 0.3$ | $41.4 \pm 0.4$ | $56.7 \pm 0.4$ |
|  | DSA | $0.10 \pm 0.02$ | $0.14 \pm 0.03$ | $1.0 \pm 0.03$ | $0.41$ | $0.29$ | $0.19$ | $26.0 \pm 0.8$ | $48.9 \pm 0.6$ | $63.0 \pm 0.4$ |
|  | KT-DSA | $0.10 \pm 0.03$ | $0.12 \pm 0.01$ | $0.18 \pm 0.03$ | $0.40$ | $0.37$ | $0.36$ | $26.0 \pm 0.8$ | $48.9 \pm 0.6$ | $63.0 \pm 0.4$ |
|  | MTT | $0.12 \pm 0.01$ | $0.15 \pm 0.01$ | $1.3 \pm 0.1$ | $0.42$ | $0.25$ | $0.12$ | $46.2 \pm 0.8$ | $65.4 \pm 0.7$ | $71.6 \pm 0.2$ |
|  | KT-MTT | $0.1 \pm 0.02$ | $0.11 \pm 0.02$ | $0.4 \pm 0.2$ | $0.42$ | $0.40$ | $0.37$ | $42.8 \pm 0.2$ | $59.8 \pm 0.2$ | $66.4 \pm 0.3$ |
|  | DATM | $0.13 \pm 0.03$ | $0.26 \pm 0.02$ | $\mathbf{1.6 \pm 0.1}$ | $0.35$ | $0.21$ | $\mathbf{0.01}$ | $46.9 \pm 0.5$ | $66.8 \pm 0.2$ | $76.1 \pm 0.3$ |
|  | KT-DATM | $0.1 \pm 0.02$ | $0.14 \pm 0.02$ | $\mathbf{0.4 \pm 0.1}$ | $0.36$ | $0.31$ | $\mathbf{0.28}$ | $43.3 \pm 0.2$ | $62.3 \pm 0.1$ | $69.2 \pm 0.2$ |
|  | RDED | $0.14 \pm 0.02$ | $0.27 \pm 0.03$ | $2.0 \pm 0.2$ | $0.02$ | $0.01$ | $0.01$ | $23.3 \pm 0.2$ | $50.2 \pm 0.3$ | $68.4 \pm 0.4$ |
|  | KT-RDED | $0.12 \pm 0.01$ | $0.18 \pm 0.03$ | $0.7 \pm 0.1$ | $0.29$ | $0.28$ | $0.28$ | $17.7 \pm 0.2$ | $42.2 \pm 0.2$ | $62.5 \pm 0.3$ |

## G COMPARISON OF TRADE-OFFS WITH DP GENERATOR

To comprehensively and fairly compare the privacy protection and data availability tradeoff of KT-DATM with other DP-generators, we conducted more comprehensive experiments on the DP-generators. For the privacy guarantee $\epsilon$, we selected values from $\{1, 5, 10, 20, 50\}$, and obtained the TPR@0.1%FPR and model accuracy under LiRA, as shown in Figure 8 . In particular, for PSG, we also conducted experiments with $\epsilon \to \infty$, i.e., without privacy protection by gradient matching noise addition.

It can be observed that as $\epsilon$ is relaxed, the data availability obtained by the DP-generator improves. For PSG, which is a dataset distillation algorithm with DP guarantees, relaxing $\epsilon$ allows it to achieve higher data availability. However, due to its outdated matching paradigm, its performance still lags

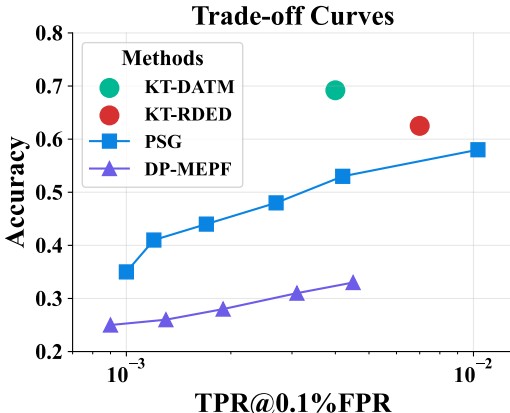

Figure 8: **Trade-off Curves of Privacy Protection and Data Availability for DP-Generators under Different** $\epsilon$**.** Under consistent protection against MIA, KT-DATM significantly outperforms DP-Generator methods in terms of data availability.

behind KT-DATM. For DP-MEPF, which only has conditional data generation under DP guarantees, the improvement in data availability is limited when relaxing $\epsilon$. However, even when achieving consistent inference attack protection, the model accuracy of KT-DATM far exceeds that of PSG and DP-MEPF.

## H  Fix-target Membership Inference Attacks on initial private data

We conduct experiments on samples both w/o and w/ our proposed KT during initialization, as displayed in Table 4 . We choose the maximum value of TPR-FPR as our threshold, and then determine whether a given sample belongs to a member based on this threshold, achieving the attack success rate. The results clearly indicate that use real data in DATM significantly leaks membership information of the initial samples. In contrast, *KT-DATM effectively preserves initial private data membership information while simultaneously maintaining generalization.*

Table 4: Perform membership inference on the **initial real samples** in TinyImageNet with `IPC = 50`.

| Method | MIA | Accuracy |
|---|---|---|
| DATM | 99.5% | 38.6% |
| KT-DATM | 54.1% | 35.2% |

