# OpenReview forum: "Privacy as a Free Lunch: Crafting Initial Distilled Datasets through the Kaleidoscope"
_ICLR.cc/2025/Conference — Submitted to ICLR 2025_

### Official Review · Reviewer_6CFr · 2024-11-02

**Soundness:** 3
**Presentation:** 3
**Contribution:** 1
**Rating:** 5
**Confidence:** 4

**Summary:**

This paper examines the privacy leakage in data distillation when real data is used for initialization. It proposes perturbing the initialization data as a method to enhance privacy protection.

**Strengths:**

This paper is well-structured and presented, with a good flow.

**Weaknesses:**

I have following concerns:

1. The concept of explicit privacy leakage introduced here seems somewhat ill-defined. Visual resemblance does not necessarily imply privacy leakage and, in some cases, could even enhance privacy. For instance, if synthesized images resemble non-sensitive images not included in the training data, they may help protect privacy, particularly when the adversary lacks access to the original dataset. In contrast, DP offers stronger and more formal privacy guarantees than this notion of explicit privacy leakage. Therefore, the motivation for introducing it is unclear to me.

2. Data distillation commonly achieves DP through random initialization (e.g., Dong et al., 2022). Your claim of additional privacy leakage, however, stems from the specific selection of real images, which is unsurprising since prior information is embedded in this selection. While this approach may affect the privacy of the chosen image, it should not impact the privacy guarantees for other images in the dataset. DP is breached in your case because DP considers the worst-case scenario. Thus, the findings in this paper appear somewhat obvious.

**Questions:**

1. What would happen if you used random initialization or added noise to the real image selection process for initialization?
2. In your case, does the adversary have access to the original dataset?

---

> ### Author Response · Authors · 2024-11-22
> **Response to Reviewer 6CFr (1/3)**
>
> Thank you for your insightful comments on the concept of **explicit privacy leakage** introduced in our paper. We appreciate your feedback and would like to provide a clarification on this topic.
>
> > W1: The concept of explicit privacy leakage introduced here seems somewhat ill-defined.
> >
>
> We thank the reviewer for the opportunity to clarify this important concept. We will separately clarify the definition of Explicit Privacy and the motivation for proposing this concept in the dataset distillation community.
>
> - **Clarify the definition of Explicit Privacy**
> In Definition 2 on line 207, we define Explicit Privacy as the similarity between the distilled dataset and **the real private data used for training** at the data level. In practice, we find that existing dataset distillation methods at high IPC directly expose the real training data used for initialization, **not just visually realistic**.
> - **Explain the motivation for defining Explicit Privacy**
>     - The motivation for introducing explicit privacy leakage is to address the limitations of existing data distillation methods, which claim to be privacy-preserving measures against membership inference attacks. However, state-of-the-art methods like trajectory matching often **use real private training data for initialization** to achieve better performance close to the original dataset. This exposure of the private data used for initialization is the key issue we aimed to address with the concept of explicit privacy leakage.
>     - Furthermore, in Proposition 1 and Remark 1, we provide a theoretical analysis showing that when the IPC is high, the randomness introduced by the **matching optimization stage** **(stage II)** is not sufficient to protect the privacy of the data used for initialization, leading to the explicit privacy leakage.
>     - It is important to note that differential privacy and explicit privacy are conceptually different. ***Differential privacy provides privacy guarantees at the random mechanism level, while explicit privacy is a data level consideration*** of whether the data will directly leak private information. The two concepts serve different purposes and can be complementary in a comprehensive privacy-preserving dataset distillation framework.

---

> ### Author Response · Authors · 2024-11-22
> **Response to Reviewer 6CFr (2/3)**
>
> > W2: This approach may affect the privacy of the chosen image, it should not impact the privacy guarantees for other images in the dataset. The findings in this paper appear somewhat obvious.
> >
>
> We appreciate the reviewer's feedback and would like to address the concerns regarding privacy leakage and the motivation behind our work.
>
> - **Privacy leakage beyond the selected images**
>     - In the random sampling without replacement initialization of Stage 1, the privacy information of the samples not selected is indeed protected. However, we also need to consider the complete process of dataset distillation, including the matching optimization process of Stage 2. During Stage 2, dataset distillation uses the real data initialized distilled dataset to perform matching optimization with the original private dataset. This process involves **accessing aggregated information from the private data**, such as gradients[1], features[2], or trajectories[3], thereby introducing privacy information of samples that were not used for initialization.
>     - As shown in Figure 3, with increasing IPC, the TPR@0.1%FPR metric of membership inference attacks shows a significant increase, indicating that the distilled dataset becomes more susceptible to membership inference attacks.
> - **We are the first work to propose that distilled datasets directly leak initialization training information**
>     - Currently, dataset distillation claims applications with privacy protection, based on the random noise initialization and distribution matching paradigm [4]. Existing works often overlook the impact of changes in distillation paradigms on privacy protection. We are the first to propose that under the current distillation paradigm, distilled datasets at high IPC  directly leak the privacy of initialization training samples, and we define this as **explicit privacy leakage**. The reason is that the randomness introduced by the matching optimization phase (line 249) decreases with increasing IPC, leading to a visual deviation of the distilled dataset from the initialized real data at low IPC (e.g., IPC=1), but **directly exposing the initialized real data at high IPC (e.g., IPC=50)**. This has significant implications for designing privacy-preserving dataset distillation methods in the future.
>
> [1] Zhao, Bo, and Hakan Bilen. "Dataset condensation with differentiable siamese augmentation." *International Conference on Machine Learning*. PMLR, 2021.
>
> [2] Zhao, Bo, and Hakan Bilen. "Dataset condensation with distribution matching." *Proceedings of the IEEE/CVF Winter Conference on Applications of Computer Vision*. 2023.
>
> [3] Cazenavette, George, et al. "Dataset distillation by matching training trajectories." *Proceedings of the IEEE/CVF Conference on Computer Vision and Pattern Recognition*. 2022.
>
> [4] Dong, Tian, Bo Zhao, and Lingjuan Lyu. "Privacy for free: How does dataset condensation help privacy?." *International Conference on Machine Learning*. PMLR, 2022.

---

> ### Author Response · Authors · 2024-11-22
> **Response to Reviewer 6CFr (3/3)**
>
> > Q1: What would happen if you used random initialization or added noise to the real image selection process for initialization?
> >
>
> We appreciate the reviewer's suggestion to use random initialization or adding noise instead of random real data sampling for initialization. Initially, dataset distillation was proposed using noise for initialization. However, due to performance limitations and optimization constraints, this approach compromised the usability of the distilled dataset.
>
> - **Random noise initialization is gradually being abandoned by existing distillation paradigms**
>     - Starting from distribution matching (DM), dataset distillation tends to use real data for initialization to achieve better aggregation of original information and higher visual realism [2, 5]. At the same time, the popular trajectory matching paradigm [3, 6], which sets a matching threshold, would fail to optimize if noise initialization were used, as the loss during the matching process would always exceed this threshold.
> - **Results obtained using noise initialization methods do not match the performance achieved with real sample initialization**
>     - We have shown the CIFAR-10 results of noise-initialized DM and PSG with $\epsilon$ approaching infinity in IPC=50. The results indicate that while these methods using noise initialization can protect the explicit privacy of the final distilled dataset, they cannot achieve the same dataset usability as KT.
>
>     | Method | TPR@0.1%FPR ($\downarrow$) | Average LPIPS Distance ($\uparrow$) | Test Accuracy ($\uparrow$) |
>     | --- | --- | --- | --- |
>     | DM-noise | 0.6±0.1 | 0.28 | 55.4±0.5 |
>     | PSG ($\epsilon\rightarrow\infty$) | 1.0±0.05 | 0.30 | 58.6±0.3 |
>     | KT-DATM | **0.4±0.1** | 0.28 | **69.2±0.2** |
> - We also explain that the multiple data augmentations of KT are a form of noise addition to real data, providing privacy protection capabilities similar to low IPC at high IPC.
>
> [5] Sun, Peng, et al. "On the diversity and realism of distilled dataset: An efficient dataset distillation paradigm." *Proceedings of the IEEE/CVF Conference on Computer Vision and Pattern Recognition*. 2024.
>
> [6] Guo, Ziyao, et al. "Towards lossless dataset distillation via difficulty-aligned trajectory matching." *arXiv preprint arXiv:2310.05773* (2023).
>
> > Q2: In your case, does the adversary have access to the original dataset?
> >
>
> In line 409, we define the attacker's access, where we release the distilled dataset and the model trained on it. Therefore, in practical attack scenarios, **the attacker does not possess the original dataset.** Subsequently, we detail the capabilities of attackers in different experiments.
>
> - **Membership Inference Attacks**
>     - The attacker can only obtain the distribution of the original dataset to sample instances for training shadow models. We follow previous work [7] and use the complete set to train shadow models.
> - **Explicit Privacy**
>     - Although the attacker does not possess the original dataset, if explicit privacy is leaked, it can lead to the attacker visually capturing that the distilled dataset itself is part of the private information.
>     - Explicit privacy allows the generator of the distilled data to effectively evaluate before releasing it, preventing the distilled dataset from directly leaking private information.
>
> [7] Carlini, Nicholas, et al. "Membership inference attacks from first principles." *2022 IEEE Symposium on Security and Privacy (SP)*. IEEE, 2022.

---

> > ### Comment · Reviewer_6CFr · 2024-11-24
> >
> > Thank you for your detailed response. I'm confused with the claim - "Differential privacy provides privacy guarantees at the random mechanism level, while explicit privacy is a data-level consideration". Privacy definitions are not inherently tied to randomness or data levels. Differential Privacy (DP) is simply a formal notion that does not prescribe specific mechanisms to achieve it.
> >
> > Your definition of explicit data privacy seems to represent a very weak interpretation of privacy. In practice, if it is possible to infer the appearance of a selected image in the input from the output, this would already violate DP (provided it is properly defined). In your case, the problem essentially reduces to determining how to perturb the input data (whether through a lossless deterministic transformation or random perturbation) to prevent inference of the input from the distilled output. This problem is well-addressed by various information-theoretic privacy approaches.
> >
> > As such, I will maintain my original score.

---

> > > ### Author Response · Authors · 2024-11-24
> > > **Response to Reviewer 6CFr (1/2)**
> > >
> > > Dear Reviewer,
> > >
> > > Thank you for your thorough feedback regarding the privacy concerns. We appreciate your expertise and would like to clarify our perspective within the context of dataset distillation:
> > >
> > > > Privacy definitions are not inherently tied to randomness or data levels. Differential Privacy (DP) is simply a formal notion that does not prescribe specific mechanisms to achieve it.
> > > >
> > > - Our understanding of DP is based on its formal definition (Definition 1 in line 205 on page 4), which specifically refers to random mechanisms $\mathcal{M}$ being $(\epsilon, \delta)$-DP.
> > >     - Differential privacy is not an attribute of the data itself, but rather an attribute of the algorithm. In other words, we can prove that an algorithm satisfies differential privacy. **If we want to prove that a dataset satisfies differential privacy, what we need to prove is that the algorithm that generates this dataset satisfies differential privacy.**
> > >     - As Dwork [1] illustrates through the coin-flipping example, "Privacy comes from the plausible deniability of any outcome," where randomness in data sampling enables such deniability and thus protects individual privacy. Randomized response allows the presentation of any outcome, including the true information, as its existence can be negated through randomness.
> > > - However, at the data level, explicit privacy **only requires the distilled dataset and the original dataset** to measure their visual similarity, **without needing to focus on the distillation process**. If explicit privacy is compromised, an attacker can obtain membership privacy by directly visualizing the distilled dataset, without needing to perform membership inference attacks.
> > > - In terms of defending against attackers, the difference lies in the following:
> > >     - **Differential Privacy**: With appropriately set privacy parameters, differential privacy can defend against membership inference attacks.
> > >     - **Explicit Privacy**: Explicit privacy protection can prevent attackers from directly obtaining membership information through visualization, making it **more suitable for scenarios where data is published**.
> > >
> > > > Your definition of explicit data privacy seems to represent a very weak interpretation of privacy. In practice, if it is possible to infer the appearance of a selected image in the input from the output, this would already violate DP (provided it is properly defined).
> > > >
> > >
> > > We acknowledge that differential privacy with appropriate privacy guarantees is sufficient to protect individual privacy. However, when discussing privacy issues **in the context of dataset distillation**, the current privacy analysis of dataset distillation is limited.
> > >
> > > - **Dataset distillation initially aims to condense datasets to improve training efficiency not for privacy preserving**: The goal of dataset distillation is to distill the knowledge from a large training dataset into a very small set of synthetic training images such that training a model on the distilled data would give a similar test performance as training one on the original dataset.
> > > - **The distilled dataset exhibits limited leakage of the original training individual data**: We measure the limited privacy leakage of individual samples through the KL divergence of neighboring datasets, thus exhibiting properties similar to differential privacy, which can **defend against membership inference attacks**, but cannot provide specific privacy parameters. Therefore, the existing dataset distillation methods do not have strict DP guarantees.
> > > - **The use of real data initialization and the existing visualization analysis have raised initial data privacy concerns about dataset distillation:** The main privacy risks currently exposed by dataset distillation is that the randomness introduced by matching optimization under high IPC is insufficient to protect **the privacy of the initialization training samples**. Therefore, in response to this phenomenon, we propose explicit privacy rather than directly using differential privacy. By using the KT plugin for explicit privacy protection, we can **minimize the impact on the task of the distilled dataset**.
> > >
> > > [1] Dwork, Cynthia, and Aaron Roth. "The algorithmic foundations of differential privacy." *Foundations and Trends® in Theoretical Computer Science* 9.3–4 (2014): 211-407.

---

> > > > ### Comment · Reviewer_6CFr · 2024-11-24
> > > >
> > > > DP stipulates the indistinguishability of posterior distributions of an output when conditioned on a sensitive data. While additive noise is a commonly used approach to achieve DP, it is not a strict requirement. In the simplest case, one can map the input to a fixed value to achieve perfect DP (albeit at the cost of no utility), which does not involve any additional randomness.
> > > >
> > > > In your paper, the notion of explicit privacy, which emphasizes perceptual differences, appears weak. An image can be encoded losslessly but still exhibit significant perceptual differences, which would clearly violate privacy while still satisfying your definition of explicit privacy.
> > > >
> > > > Information-theoretic privacy frameworks, such as "Generative Adversarial Privacy," offer a more robust approach. These frameworks optimize the trade-off between privacy and utility (which, in your case, corresponds to data distillation), providing stronger privacy guarantees and likely achieving better utility outcomes.

---

> > > > > ### Author Response · Authors · 2024-11-24
> > > > > **Response to Reviewer 6CFr**
> > > > >
> > > > > Thank you for your thoughtful comments and for continuing the discussion on privacy protection in the context of dataset distillation. We appreciate your insights and the opportunity to clarify our approach.
> > > > >
> > > > > - **Dataset Distillation Context:** Dataset distillation involves compressing large original datasets into much smaller distilled datasets while ensuring that the utility of the distilled datasets remains comparable to the original datasets.
> > > > > - **Current pure privacy protection methods are not feasible in this context** because they lack an information aggregation process, which is essential for maintaining data utility in the distilled datasets.
> > > > >
> > > > > > DP stipulates the indistinguishability of posterior distributions of an output when conditioned on a sensitive data. While additive noise is a commonly used approach to achieve DP, it is not a strict requirement. In the simplest case, one can map the input to a fixed value to achieve perfect DP (albeit at the cost of no utility), which does not involve any additional randomness.
> > > > > >
> > > > > - **Regarding DP:** We acknowledge that Differential Privacy is a robust framework for privacy protection. However, in the context of dataset distillation, the example you provided—mapping the input to a fixed value to achieve perfect DP—is indeed impractical.
> > > > >     - The goal of dataset distillation is to **compress a large dataset (e.g., 50,000 training images from CIFAR-10) into a much smaller distilled dataset (e.g., 10, 100, or 500 images)** while maintaining comparable utility.
> > > > >     - The current DP paradigms struggle to achieve data utility under limited IPC conditions. In contrast, our KT plugin demonstrates comparable membership inference attack protection to other differential privacy guarantees methods (e.g., $ε=10$ for PSG and DP-MEPF) while significantly outperforming these methods in terms of data utility.
> > > > >
> > > > > > In your paper, the notion of explicit privacy, which emphasizes perceptual differences, appears weak. An image can be encoded losslessly but still exhibit significant perceptual differences, which would clearly violate privacy while still satisfying your definition of explicit privacy.
> > > > > >
> > > > > - **Regarding Explicit Privacy:** We understand your concern about the notion of explicit privacy in our paper.
> > > > >     - The explicit privacy we propose is specifically tailored to the context of dataset distillation, where the distilled dataset inherently leaks privacy information from the initialization samples. The dataset distillation community has not yet fully recognized the complete leakage of initialization privacy samples in the distilled datasets, and it is crucial to raise awareness on this issue.
> > > > >     - Our KT plugin effectively protects explicit privacy without violating the privacy concerns you mentioned. We have demonstrated through KL divergence analysis between neighboring datasets that the leakage of individual privacy is limited. Our experimental results show strong defense against membership inference attacks.
> > > > >
> > > > > > Information-theoretic privacy frameworks, such as "Generative Adversarial Privacy," offer a more robust approach. These frameworks optimize the trade-off between privacy and utility (which, in your case, corresponds to data distillation), providing stronger privacy guarantees and likely achieving better utility outcomes.
> > > > > >
> > > > >
> > > > > We agree that information-theoretic privacy frameworks like Generative Adversarial Privacy (GAP) offer a robust approach to privacy protection. However, there are two key differences when comparing GAP to our approach in the context of dataset distillation:
> > > > >
> > > > > - **Distilled Dataset as Privacy Variable:** In dataset distillation, the distilled dataset itself is the privacy variable. There is no effective public variable to which perturbations can be applied. Our goal is to ensure that the distilled dataset does not reveal membership information of the original dataset, rather than other privacy features.
> > > > > - **Data Utility in Dataset Distillation:** GAP-like privacy protection methods do not achieve data utility in the dataset distillation scenario. For example, compressing 50,000 images into 500 images would significantly reduce data utility because these methods lack an information aggregation process that matches the original dataset's information.
> > > > >
> > > > > In summary, we encourage the reviewer to consider privacy protection from the perspective of dataset distillation. While DP is an effective privacy protection method, its current integration with dataset distillation has not achieved a good trade-off. We hope that the dataset distillation community will focus on explicit privacy leakage in the future, contributing to more stringent privacy protection on DP.
> > > > >
> > > > > Thank you again for your valuable feedback and for the opportunity to address these points. We look forward to further discussion and hope that our responses clarify our approach.

---

> > > > > > ### Comment · Reviewer_6CFr · 2024-11-26
> > > > > >
> > > > > > 1. The example of perfect DP is just a showcase that DP is not necessarily tied to randomness, contrary to your claim in the response. My primary concern, however, is that the notion of explicit privacy is inherently problematic as a privacy criterion, regardless of the specific tasks or models to which it is applied. It risks providing a false sense of security while neglecting critical privacy concerns.
> > > > > >
> > > > > > 2. There are many established methods to balance privacy and utility in machine learning tasks. General frameworks, such as those developed in Shokri's works, provide rigorous and flexible mechanisms for this purpose. I do not see why data distillation as a utility measure is different from these frameworks or why it requires an alternative privacy notion.
> > > > > >
> > > > > > 3. Your analysis of DP appears flawed. Augmented samples are generated from the original data, creating a strong correlation between the two. However, in your analysis, these inputs are treated as independent samples, which undermines the validity of privacy protection of original data.

---

> > > > > > > ### Comment · Reviewer_6CFr · 2024-11-26
> > > > > > >
> > > > > > > I also completely agree with the comments from Reviewer 4Amd. There is a fundamental flaw in the method proposed for achieving Differential Privacy (DP).

---

> > > > > > > > ### Author Response · Authors · 2024-11-26
> > > > > > > > **Response to Reviewer 6CFr (1/2)**
> > > > > > > >
> > > > > > > > Thank you for your insightful comments and constructive feedback. We appreciate the opportunity to clarify and address your concerns.
> > > > > > > >
> > > > > > > > We understand that the misunderstanding regarding explicit privacy persists. To clarify, explicit privacy is not an isolated privacy criterion. It is a necessary privacy protection measure in the context of distilled dataset release.
> > > > > > > >
> > > > > > > > > The example of perfect DP is just a showcase that DP is not necessarily tied to randomness, contrary to your claim in the response.
> > > > > > > > >
> > > > > > > >
> > > > > > > > We understand that there might have been a misunderstanding regarding the distinction between DP and explicit privacy in our previous response.
> > > > > > > >
> > > > > > > > - We emphasize that differential privacy is fundamentally an attribute of the algorithm rather than the data itself. In your example, the algorithm that maps inputs to a fixed value is indeed DP.
> > > > > > > > - Explicit privacy, on the other hand, is a measure of privacy at the data level, **independent of the algorithm**.
> > > > > > > > - Additionally, randomness is necessary from the perspective of data utility and aligns with Dwork's definition of differential privacy [1].
> > > > > > > >
> > > > > > > > [1] Dwork, Cynthia, and Aaron Roth. "The algorithmic foundations of differential privacy." *Foundations and Trends® in Theoretical Computer Science* 9.3–4 (2014): 211-407.
> > > > > > > >
> > > > > > > > > My primary concern, however, is that the notion of explicit privacy is inherently problematic as a privacy criterion, regardless of the specific tasks or models to which it is applied. It risks providing a false sense of security while neglecting critical privacy concerns.
> > > > > > > > >
> > > > > > > >
> > > > > > > > We acknowledge your concern about the potential risks associated with explicit privacy. However, it is important to note that explicit privacy is not used in isolation in our experiments. We employ both Membership Inference Attacks and data visualization to assess the privacy of the distilled dataset, not solely relying on explicit privacy.
> > > > > > > >
> > > > > > > > - Given that our privacy protection scenario involves the release of distilled datasets, which differs from the typical model release in machine learning privacy, it is crucial to consider explicit privacy.
> > > > > > > >     - Attackers may directly visualize the image data, potentially revealing sensitive information about the initialization samples. Therefore, explicit privacy evaluation is necessary to address this direct risk, which does not require additional attacks from the adversary.
> > > > > > > >     - To achieve robust data privacy protection, addressing this risk is essential, and we highlight the overlooked issue of initialization sample privacy in the dataset distillation community.
> > > > > > > >
> > > > > > > > > I do not see why data distillation as a utility measure is different from these frameworks or why it requires an alternative privacy notion.
> > > > > > > > >
> > > > > > > >
> > > > > > > > Data distillation requires high data utility, as it aims to maintain performance comparable to the original dataset while **using only 1% or even less of the original data size**. Existing DP methods struggle to achieve such high utility with such small datasets.
> > > > > > > >
> > > > > > > > Shokri's work introduces the concept of Data Minimization [2], which aligns with the natural constraints of data distillation in terms of data size. However, the data reduction proposed by Data Minimization does not meet the requirements of dataset distillation, which primarily aims to enhance training efficiency by obtaining a very small dataset (1% or even less of the original data size). In the context of dataset distillation, mainstream distillation methods expose initialization samples privacy leakage.
> > > > > > > >
> > > > > > > > Given the data release scenario, where attackers can directly access and visualize the data, explicit privacy measures are essential before releasing the distilled dataset.
> > > > > > > >
> > > > > > > > [2] Ganesh, Prakhar et al. “The Data Minimization Principle in Machine Learning.” *ArXiv* abs/2405.19471 (2024): n. pag.
> > > > > > > >
> > > > > > > > > Augmented samples are generated from the original data, creating a strong correlation between the two. However, in your analysis, these inputs are treated as independent samples, which undermines the validity of privacy protection of original data.
> > > > > > > > >
> > > > > > > >
> > > > > > > > We clarify that Lemma 1 (in line 270 on page 6) in our paper explicitly describes the relationship between the distilled dataset and the original dataset. Theorem 1 is subsequently proven based on Lemma 1, demonstrating that the distilled dataset's privacy leakage concerning the original dataset's samples is bounded. This approach ensures the validity of our privacy protection analysis.

---

> > > > > > > > ### Author Response · Authors · 2024-11-26
> > > > > > > > **Response to Reviewer 6CFr (2/2)**
> > > > > > > >
> > > > > > > > > I also completely agree with the comments from Reviewer 4Amd. There is a fundamental flaw in the method proposed for achieving Differential Privacy (DP).
> > > > > > > > >
> > > > > > > >
> > > > > > > > We would like to clarify that our work does not claim that the KT plugin's dataset distillation achieves DP. Instead, we analyze the upper bound of individual data leakage through the KL divergence between neighboring datasets, which exhibits properties similar to DP.
> > > > > > > >
> > > > > > > > - If you refer to Proposition 1 as the "fundamental flaw," we have already addressed this in our responses to Reviewer 4Amd and rmpC. Proposition 1 is a privacy analysis based on random true sampling, which prompted our focus on initialization sample privacy. It is not an error but rather a motivation to highlight the importance of explicit privacy. **Reviewer rmpC understood this and improved our score, while Reviewer 4Amd did not further question Proposition 1.**
> > > > > > > >
> > > > > > > > Once again, thank you for your valuable feedback. We hope these clarifications address your concerns and strengthen the validity of our work.

---

> > > ### Author Response · Authors · 2024-11-24
> > > **Response to Reviewer 6CFr (2/2)**
> > >
> > > > In your case, the problem essentially reduces to determining how to perturb the input data (whether through a lossless deterministic transformation or random perturbation) to prevent inference of the input from the distilled output. This problem is well-addressed by various information-theoretic privacy approaches.
> > > >
> > > - We believe there may exist a misunderstanding regarding our objective. **We cannot deviate from the primary task of dataset distillation.** Differential privacy is a more stringent privacy guarantee for existing dataset distillation methods.
> > >     - Our primary goal is to **maintain the utility of distilled datasets** while protecting their explicit privacy against direct privacy information extraction. Our proposed KT plugin achieves this by both safeguarding explicit privacy and enhancing resistance to membership inference attacks, achieving the current best trade-off.
> > >     - While we acknowledge that DP-guaranteed distillation algorithms can thoroughly protect individual data privacy, they are not well-suited for the current real-data initialization paradigm, significantly reducing data utility (see lines 1016-1048 on page 19).
> > >     - **We are the first work to propose that distilled datasets directly leak initialization training information**: It is important to draw the attention of the dataset distillation community to the serious issue of initialization data leakage.

---

> ### Author Response · Authors · 2024-12-02
> **A Kind Reminder to Reviewer 6CFr**
>
> Dear Reviewer 6CFr,
>
> Thank you once again for your insightful feedback on our submission. We would like to remind you that the discussion period is concluding. To facilitate your review, we have provided a concise summary below, outlining our responses to each of your concerns:
>
> - **Privacy leakage in data release scenarios is not guaranteed by differential privacy**: Differential privacy protects membership identity, whereas explicit privacy safeguards against privacy leakage in data release (as illustrated in the new PDF, Figure 1). Using your example of DP with fixed values, if the fixed value is sensitive information, it does not violate DP, but an attacker can directly obtain this private information. **The security of distilled data release is ensured through evaluations of both explicit privacy and membership inference attacks.**
> - The distilled dataset needs to maintain performance at a much smaller scale than the original dataset. However, the DP methods currently cannot meet this availability standard, necessitating the use of distillation techniques to ensure the utility of the concentrated data.
> - **A Comprehensive Revision**: We have revised the title and the main text to avoid over-claiming, added scene images and descriptions to better understand the distillation data scenario, clarified the relationship with differential privacy, and reorganized the contributions of this paper.
>
> More detailed information can be found in the general response and the revised PDF. We are grateful for your insightful comments and are eager to confirm whether our responses have adequately addressed your concerns. We look forward to any additional input you may provide.
>
> Warm regards,
>
> The Authors of Submission 10266.

---

### Official Review · Reviewer_4Amd · 2024-11-02

**Soundness:** 1
**Presentation:** 2
**Contribution:** 1
**Rating:** 1
**Confidence:** 5

**Summary:**

This paper investigates the issue of privacy leakage in the process of dataset distillation. The authors point out that existing distillation methods produce datasets that closely resemble real data, leading to significant privacy leakage, termed as explicit privacy leakage. The paper analyzes how a high IPC (images per class) can weaken the protection of both differential privacy and explicit privacy.

To address this issue, the authors propose a module called "Kaleidoscopic Transformation" (KT), which applies strong perturbations to real data during the initialization phase to protect privacy. Experimental results show that this method ensures both differential privacy and explicit privacy while maintaining the generalization performance of the distilled data.

In summary, the contributions of the paper include:

1. Identifying and analyzing the problem of explicit privacy leakage caused by high IPC.

2. Theoretically demonstrating that using real data during the initialization phase leads to privacy leakage.

3. Proposing and validating a module that enhances privacy protection without sacrificing performance.

**Strengths:**

# Originality
The Kaleidoscopic Transformation (KT) module is a creative approach to enhance privacy by introducing strong perturbations during data initialization, which is an original contribution to the field.
# Clarity
The paper clearly articulates the problem of explicit privacy leakage and its implications, making it accessible to a broad audience.
The description of the KT module and its integration into existing distillation methods is clearly presented, allowing others to replicate the work.

**Weaknesses:**

I believe the critical flaw in this paper lies in its attempt to ensure differential privacy through random sampling of datasets, as described in Proposition 1. While sampling can introduce randomness, the problem is that the algorithm fully exposes the privacy of the selected samples without additional protection. This does not meet the requirements of differential privacy. Specifically, refer to the second paragraph on page 18 of Dwork's privacy book [1] . For convenience, I will quote that section: "Typically we are interested in values of (\delta) that are less than the inverse of any polynomial in the size of the database. In particular, values of (\delta) on the order of (1/||x||_1) are very dangerous: they permit 'preserving privacy' by publishing the complete records of a small number of database participants." ( ||x||_1 refers to the size of the dataset, in the is paper, it should be |T| ) In the authors' paper, Proposition 1 describes exactly this situation: completely sacrificing the privacy of the sampled participants. Therefore, I believe that the proof based on Proposition 1 provides meaningless differential privacy protection (\delta=|s|/|T|). In fact, differential privacy cannot be achieved solely through sampling; it still requires the addition of noise, such as Gaussian noise. When Gaussian noise is added, sampling can enhance privacy to some extent, as in the DPSGD algorithm, which still adds isotropic Gaussian noise to the gradients obtained from sampling. I consider this a fatal flaw in the paper.

[1] Dwork, C., & Roth, A. (2014). The algorithmic foundations of differential privacy. Foundations and Trends® in Theoretical Computer Science, 9(3–4), 211-407.

**Questions:**

I think the question above is critical.

Some of the citation is ?, please check the latex code.

---

> ### Author Response · Authors · 2024-11-22
> **Response to Reviewer 4Amd (1/2)**
>
> > Proposition 1 provides meaningless differential privacy protection ($\delta=\frac{|\mathcal{S}|}{|\mathcal{T}|}$).
> >
>
> Thank you for your feedback. We assume there may be a misunderstanding about the role of random sampling in our method that we'd like to clarify. We will interpret the significance of Proposition 1 from the perspectives of what, how, and why. Finally, we will address the reviewer's concern about $\delta$ with targeted experiments.
>
> 1. **Comprehensive privacy analysis of the distilled dataset:** We would like to emphasize that our privacy analysis of the distilled dataset is a holistic process that involves **two stages**, as defined in **Line 182** of our paper. Proposition 1 pertains solely to the privacy analysis of the initial random sampling phase. The privacy guarantees of the distilled dataset are further analyzed in Theorem 1 and Theorem 2, where we use the Kullback-Leibler divergence between neighboring datasets to measure the privacy of the distilled dataset. Our findings indicate that the distilled dataset exhibits properties akin to differential privacy.
> 2. **Clarification on the Value of** $\delta$**, which motivates us to focus on the privacy leakage of initialization samples**
>     1. We acknowledge the concern raised regarding the value of $\delta$ obtained in Proposition 1, which is $\delta=\frac{|\mathcal{S}|}{|\mathcal{T}|}$. This value indeed exceeds the threshold of $\frac{1}{|\mathcal{T}|}$ as described in Dwork's privacy book. However, it is important to note that we do not directly use the randomly sampled subset of real data as the distilled dataset, as this would indeed compromise the privacy of the sampled data.
>     2. In Proposition 1 and Remark 1, we explicitly state that $|\mathcal{S}|/|\mathcal{T}|$ represents **the proportion of real privacy data selected for initialization**. This proportion indicates that the privacy of these initial samples cannot be fully protected through the fluctuations in the matching optimization process. This realization **motivated us to introduce the concept of explicit privacy and the KT plugin to safeguard the privacy of these initialization samples**.
>     3. Dwork's assertion that values of $\delta$ on the order of $1/\Vert x\Vert_1$ are dangerous stems from the **"just a few" philosophy**, refering to the last paragraph on page 9 of Dwork's privacy book [1] . The book states ”For a single data set, ‘just a few’ privacy can be achieved by randomly selecting a subset of rows and releasing them in their entirety”, where the privacy of a very small number of samples is directly compromised. In our work, the introduction of **explicit privacy** and the **KT plugin** is precisely aimed at **addressing this issue**, **rather than being a weakness or flaw**. The dataset distillation process introduces perturbations from other samples during the matching optimization phase, but when the IPC is high, these perturbations are insufficient to protect explicit privacy, as illustrated in Figure 1. The KT plugin, therefore, applies perturbations during the initialization phase to protect explicit privacy while reducing the KL divergence between neighboring datasets.
> 3. **In addition to revealing that random sampling can lead to privacy leakage of initialization samples, Proposition 1 effectively protects non-initialization samples:** Proposition 1 is merely the first stage of the dataset distillation process. We encourage the reviewer to consider the **entire framework of dataset distillation**. Proposition 1 effectively compresses the sample space, thereby significantly protecting the privacy of the majority of the data and **effectively defending against membership inference attacks**. For the privacy of the initialization samples, we provide protection through the matching optimization process and the KT plugin.

---

> ### Author Response · Authors · 2024-11-22
> **Response to Reviewer 4Amd (2/2)**
>
> 4. **Experimental Validation of Initialization Sample Privacy:** We have conducted experiments to address the concerns regarding the **privacy of the initialization samples**, as raised by both you and Reviewer rmpC. **In Table 4 at Line 1053** of our paper, we specify the initialization samples for **a fix target membership inference attack** [2].
>     - Notably, the success rate of membership inference attacks on initialization training samples decreased from 99.5% to 54.1%. The experimental results demonstrate that the introduction of the KT plugin effectively makes the privacy information of the initialization samples less susceptible to membership inference attacks, thereby protecting their explicit privacy and effectively defending against such attacks.
>
> We hope these clarifications address your concerns and provide a more comprehensive understanding of our approach to privacy protection in dataset distillation. Thank you once again for your valuable feedback.
>
> [1] Dwork, Cynthia, and Aaron Roth. "The algorithmic foundations of differential privacy." *Foundations and Trends® in Theoretical Computer Science* 9.3–4 (2014): 211-407.
>
> [2] Ye, Jiayuan, et al. "Enhanced membership inference attacks against machine learning models." *Proceedings of the 2022 ACM SIGSAC Conference on Computer and Communications Security*. 2022.

---

> > ### Comment · Reviewer_4Amd · 2024-11-25
> >
> > I have reviewed the author's response as well as the discussions between the author and other reviewers. To the best of my understanding, privacy protection can be categorized into two types: those with theoretical guarantees, such as differential privacy (DP), and those validated through experiments.
> >
> > Theoretical guarantees for privacy require providing worst-case loss bounds, as in DP. This paper does not achieve that. I believe the method proposed in this paper can only be validated experimentally. However, this is risky, as we cannot be sure whether the authors have implemented sufficiently strong attacks or whether stronger attacks may emerge in the future.
> >
> > Moreover, experimental validation is data-dependent, and we cannot determine whether the privacy protection would remain effective with different datasets. For example, consider dataset A consisting of n pure white images and dataset B consisting of n-1 white images and one pure black image. These datasets satisfy the definition of neighboring datasets in differential privacy. However, under the proposed method, I believe simple data augmentations (as shown in Figure 2) cannot make the black image indistinguishable. Therefore, I find the privacy protection provided by this paper to be very weak and unreliable. Reviewers rmpC and 6CFr have expressed similar concerns.
> >
> > Additionally, since the proposed privacy protection method is weak, I believe the title and writing of this paper tend to overclaim, which could provoke negative reactions from the privacy community. I suggest the authors revise their work in future submissions.
> >
> > In conclusion, I will maintain my recommendation to reject this paper.

---

> > > ### Author Response · Authors · 2024-11-25
> > > **Response to Reviewer 4Amd (1/2)**
> > >
> > > First and foremost, we would like to express our sincere gratitude for the opportunity to further discuss our work and address the concerns raised by the reviewers. Our research focuses on the task of dataset distillation, which involves compressing large datasets (e.g., 50,000 images) into much smaller datasets (e.g., 10, 100, or 500 images) while maintaining the performance of models trained on these distilled datasets comparable to those trained on the original datasets. In this context, we have identified significant privacy benefits, particularly in terms of explicit privacy protection and enhanced resilience against membership inference attacks.
> > >
> > > > Theoretical guarantees for privacy require providing worst-case loss bounds, as in DP. This paper does not achieve that. I believe the method proposed in this paper can only be validated experimentally.
> > > >
> > >
> > > We have **theoretically analyzed** the individual data leakage bounds using the Kullback-Leibler divergence between adjacent datasets, both without and with the KT plugin, as stated in Theorem 1 (Line 292, Page 6) and Theorem 2 (Line 347, Page 7). These theorems demonstrate that the distilled dataset effectively resists membership inference attacks, and the KT plugin further tightens these bounds, enhancing protection.
> > >
> > > > However, this is risky, as we cannot be sure whether the authors have implemented sufficiently strong attacks or whether stronger attacks may emerge in the future.
> > > >
> > >
> > > Regarding the reviewer's concern about the risk of strong attacks, we explicitly mention the attack methods we employed. For membership inference attacks, we used LiRA [1], a proven effective attack method, in line with previous work evaluating generated data[2, 3]. We also designed a fair evaluation framework (Line 948, Page 18) to ensure a fair comparison between the original and distilled datasets.
> > >
> > > The effectiveness of our attacks is evident in Table 1 (Line 378, Page 8).
> > >
> > > For example on TinyImagenet:
> > >
> > > - **Full dataset**: The TPR@0.1%FPR for the full dataset reaches 17.3±0.5%. This demonstrates the effectiveness of our attack method.
> > > - **Distilled dataset with IPC=50**:
> > >     - DATM with IPC=50, reduces this to 2.4±0.1%. This reduction is attributed to the distilled dataset's ability to shrink the sample space and effectively match the optimization of the distilled dataset, thereby achieving significant resistance to MIA.
> > >     - KT-DATM with IPC=50, the TPR@0.1%FPR further decreases to 0.5±0.2%. As analyzed in Theorem 2, the introduction of the KT plugin not only protects the explicit privacy of the distilled dataset but also tightens the upper bound on individual privacy leakage, further enhancing the resistance to such attacks.
> > >
> > > As of our current knowledge, there are no stronger privacy attacks specifically targeting distilled datasets, which could be a topic for future research.
> > >
> > > > I find the privacy protection provided by this paper to be very weak and unreliable.
> > > >
> > >
> > > The reviewer raises a concern about the data-dependency of experimental validation and whether the privacy protection would remain effective with different datasets.
> > >
> > > - We believe that the extreme example provided **is not representative** of our target large datasets to distill, which consist of complex and diverse images.
> > > - The KT plugin shown in Figure 2 is just one step in our process, followed by a matching optimization phase where the distilled dataset is optimized based on the aggregate information of the original dataset, introducing further perturbations.
> > > - For outliers in the selected initialization images, the matching optimization phase ensures that the distilled dataset captures the corresponding attributes of the original large dataset, applying greater perturbations to such outliers.
> > > - Regarding explicit privacy, we emphasize that **it is a significant privacy risk identified in the distilled datasets.** Attackers can visually inspect the data to obtain privacy information about the initialization samples, which poses a significant privacy risk in the data generation and dataset distillation communities.
> > >
> > > [1] Carlini, Nicholas, et al. "Membership inference attacks from first principles." *2022 IEEE Symposium on Security and Privacy (SP)*. IEEE, 2022.
> > >
> > > [2] Dong, Tian, Bo Zhao, and Lingjuan Lyu. "Privacy for free: How does dataset condensation help privacy?." *International Conference on Machine Learning*. PMLR, 2022.
> > >
> > > [3] Yuan, Jianhao, et al. "Real-fake: Effective training data synthesis through distribution matching." *arXiv preprint arXiv:2310.10402* (2023).

---

> > > ### Author Response · Authors · 2024-11-25
> > > **Response to Reviewer 4Amd (2/2)**
> > >
> > > > Additionally, since the proposed privacy protection method is weak, I believe the title and writing of this paper tend to overclaim, which could provoke negative reactions from the privacy community. I suggest the authors revise their work in future submissions.
> > > >
> > >
> > > We understand the reviewer's concern about the potential overclaiming in our title and writing. We clarify that the term "free" refers to the fact that we are still performing dataset distillation, with explicit privacy protection and enhanced resistance to membership inference attacks as additional benefits. Our KT plugin combined with the dataset distillation framework allows us to achieve comparable data utility with a much smaller distilled dataset. We will consider clarifying this in the main text to avoid any ambiguity.
> > >
> > > We believe that our work provides valuable contributions to the field of dataset distillation with enhanced privacy protection. We hope that the responses above address the reviewer's concerns and provide a clearer understanding of our approach and its benefits.

---

> ### Author Response · Authors · 2024-12-02
> **A Kind Reminder to Reviewer 4Amd**
>
> Dear Reviewer 4Amd,
>
> Thank you once again for your insightful feedback on our submission. We would like to remind you that the discussion period is concluding. To facilitate your review, we have provided a concise summary below, outlining our responses to each of your concerns:
>
> - **Proposition 1 is the motivation for our proposed explicit privacy, not a flaw.**
> - **Explicit Privacy and Differential Privacy**: Differential privacy protects membership identity, whereas explicit privacy safeguards against privacy leakage in data release (as illustrated in the new PDF, Figure 1). Theoretically, we analyze the leakage of individual samples through KL divergence. In our experiments, we assess membership inference attacks and explicit privacy on distilled data, comparing the original dataset and KT plugin results. The KT plugin significantly enhances explicit privacy and defense against inference attacks while preserving original distillation effects.
> - **A Comprehensive Revision**: We have revised the title and the main text to avoid overclaiming, added scene images and descriptions to better understand the distillation data scenario, clarified the relationship with differential privacy, and reorganized the contributions of this paper.
>
> More detailed information can be found in the general response and the revised PDF. We are grateful for your insightful comments and are eager to confirm whether our responses have adequately addressed your concerns. We look forward to any additional input you may provide.
>
> Warm regards,
>
> The Authors of Submission 10266.

---

> > ### Comment · Reviewer_4Amd · 2024-12-03
> >
> > Thamks, I read your rebuttal and think your revision cannot solve my concerns and the privacy protection in this paper is weak and have fundamental flaw. I think reviewers 6CFr and rmpC have the same conclusion. I believe this paper should be rejected.

---

### Official Review · Reviewer_rmpC · 2024-11-02

**Soundness:** 2
**Presentation:** 3
**Contribution:** 2
**Rating:** 5
**Confidence:** 3

**Summary:**

This paper studies privacy guarantees in data distillation methods. It introduces the concept of explicit privacy, which considers how similar the generated data is to the original data, and then examines the extent to which this privacy metric is preserved in data distillation methods. The paper also analyzes the relationship between privacy guarantees in data distillation and differential privacy.

**Strengths:**

- S1: The paper is well-organized and easy to follow.
- S2: The paper identifies that distilled datasets produced by state-of-the-art distillation methods strongly resemble real data, indicating a significant risk of privacy leakage. The problem addressed is interesting and relevant.
- S3: Experimental results provide various comparisons with existing methods.

**Weaknesses:**

- W1: The discussion of differential privacy achieved through random sampling in data distillation appears problematic. It closely resembles the concept of privacy amplification via subsampling. Privacy amplification through subsampling assumes that data has already been privatized to a certain extent, and this privacy level is then amplified by subsampling. However, Proposition 1 assumes that none of the data has been privatized to ensure differential privacy prior to subsampling. Consequently, the subsampled data do not satisfy any differential privacy guarantees. Please clarify how the proposed approach differs from privacy amplification via sub-sampling. Also please provide a more rigorous justification for why the random sampling method in data distillation provides differential privacy guarantees without prior privatization.
- W2: Even if Proposition 1 were applicable, it seems impractical because $\delta$ would need to be less than or equal to $n^{-1}$, which implies $S = 1$. This would mean that the initialization process in data distillation is restricted to sampling only one record from the entire dataset of size T. Such a limitation may be undesirable for data distillation methods. This paper needs to clarify how the proposed method can be practically implemented given such constraints. If there are any relaxations or modifications that could make it more feasible for real-world data distillation scenarios, including these things in this paper might be helpful.
- W3: The comparison with differentially private generators appears to be unfair. While these DP generators are trained under $\epsilon = 10$, the proposed method does not assume the same privacy level. For a fair comparison, the authors should evaluate performance at an equivalent privacy level. A possible suggestion is that the authors conduct additional experiments where their method is constrained to provide equivalent privacy guarantees as the DP generators and to provide a more detailed discussion of how privacy levels can be meaningfully compared between the proposed approach and existing DP methods.

Overall, the analytical study on differential privacy in data distillation appears to have several significant issues. If there are any misunderstandings in my interpretation, please clarify them.

**Questions:**

For W1, W2, and W3, if there are any misunderstandings in my interpretation, please clarify them.

Q1: Please clarify how the proposed approach differs from privacy amplification via sub-sampling. Also, please provide a more rigorous justification for why the random sampling method in data distillation provides differential privacy guarantees without prior privatization.

Q2: How can the proposed method be practically implemented, given the constraints described in W2? In addition, if there are any relaxations or modifications that could make it more feasible for real-world data distillation scenarios, including these things in this paper might be helpful.

---

> ### Author Response · Authors · 2024-11-22
> **Response to Reviewer rmpC (1/3)**
>
> We would like to thank you for your time spent reviewing our paper and for raising questions about **the validity of Proposition 1** and **the fairness of the comparison with the DP-generator**. Below, I will address these concerns separately.
>
> > W1&Q1: Difference between random sampling for initialization and privacy amplification via sub-sampling. Why does the random sampling method in data distillation provide differential privacy guarantees without prior privatization?
> >
>
> To address your question comprehensively, we will first explain the mechanism of random sampling for initialization in our method, then demonstrate the correctness of its privacy guarantees through theoretical analysis, and finally elucidate its key distinctions from privacy amplification via sub-sampling.
>
> - **Random sampling for initialization**
>
>     As mentioned on line 182, we divide the data distillation process into **two stages**: the initialization stage and the matching optimization stage. Proposition 1 analyzes only the first stage, where samples are randomly selected to initialize the distilled dataset, and concludes that it satisfies $(\ln\frac{|\mathcal{T}|+1}{|\mathcal{T}|+1-|\mathcal{S}|},\frac{|\mathcal{S}|}{|\mathcal{T}|})$-DP. However, to analyze the privacy protection of the entire distillation process, one must refer to Theorem 1 and Theorem 2 after the introduction of KT, particularly how the KT plugin and subsequent optimization process protect the private training data corresponding to the initialized distillation samples.
>
> - **Proposition 1 is correct and it justifies the limitation that dataset distillation methods with real data initialization would leak the privacy of corresponding initialized real samples.**
>     - We provide complete proof of Proposition 1 **in lines 792 on page 15 (Appendix B)**, demonstrating the differential privacy properties of random selection according to the definition of differential privacy.
>     - We can also understand this from the perspectives of $\epsilon$ and $\delta$. $\epsilon$ represents the scaling value of the probability of obtaining the same subset, while $\delta$  represents the probability of failing to obtain the same subset, i.e., the probability of privacy protection failure. Our obtained $\delta=\frac{|\mathcal{S}|}{|\mathcal{T}|}$, which is precisely the probability of selecting the differing sample in adjacent datasets. If the differing sample is selected as a subset member, privacy protection fails.
>     - However, $\delta=\frac{|\mathcal{S}|}{|\mathcal{T}|}$ can also be interpreted as leading to the privacy leakage of the selected samples, a point raised in your W2, so that we propose KT plugin. We will further address this in W2.
> - **Difference from privacy amplification via sub-sampling**
>
>     The distinction between our approach and privacy amplification via sub-sampling lies in their ***fundamental mechanisms and objectives***:
>
>     - Our random selection without replacement serves as an integral component of the data initialization process, where Proposition 1 establishes the privacy guarantees specifically for the unselected samples. While this mechanism may potentially expose the privacy of initialized samples (as we will address in the subsequent response), its primary purpose is to establish baseline privacy properties during the initialization phase.
>     - In contrast, privacy amplification via sub-sampling operates as a post-processing technique that enhances existing privacy protection mechanisms. It achieves this by reducing the effective privacy budget through random sampling, thereby strengthening the overall privacy guarantees of the original mechanism.

---

> ### Author Response · Authors · 2024-11-22
> **Response to Reviewer rmpC (2/3)**
>
> > W2&Q2:  $\delta$ would need to be less than or equal to $n^{-1}$. Any relaxations or modifications that could make it more feasible?
> >
>
> We would like to emphasize that we use Proposition 1 to motivate the **explicit privacy leakage** problem and the design of our KT plugin, and our proposal consists of **two stages: (1) random training data sampling for initialization, and (2) matching optimization with randomness**.
>
> - **Proposition 1 reveals training data privacy used for initialization**
>     - In Proposition 1, we obtain $\delta=\frac{|\mathcal{S}|}{|\mathcal{T}|}$. This value is greater than the differential privacy requirement of less than $\frac{1}{|\mathcal{T}|}$, indicating that the privacy of the $|\mathcal{S}|$samples used for initialization is compromised. **This is precisely the motivation for our explicit privacy leakage.**
>     - Based on our analysis in Remark 1, we believe that even after the matching optimization process (Theorem 1), the samples used for initialization in high IPC distillation methods still face significant privacy leakage risks.
>     - Therefore, **we propose the KT plugin to protect the privacy of these initialized samples**. This should not be seen as a weakness but rather as the privacy risk inherent in dataset distillation methods that use real data for initialization.
> - **Privacy analysis of dataset distillation in two stages yields privacy protection properties of the final distilled dataset**
>     - It is important to note that Proposition 1 pertains only to the first stage of dataset distillation, not the final optimized distilled dataset. Proposition 1 not only reveals the leakage of initialization training samples but also demonstrates the effective protection of training samples not used for initialization, thereby effectively defending against membership inference attacks.
>     - The complete privacy analysis of dataset distillation requires reference to Theorem 1 and Theorem 2 after the introduction of the KT plugin. In the second matching optimization stage, we analyze **the properties similar to differential privacy using the KL divergence of adjacent datasets**. At the same time, the optimization brought by matching is considered as privacy protection for initialization training samples. As seen in Figure 1, the information of the initialized data is protected at low IPC. We emphasize the use of the KT plugin to address the issue of explicit privacy leakage at high IPC.
> - **Validate the above discussion with experiments**
>     - Existing dataset distillation methods do not consider the explicit privacy leakage risks associated with high IPC. The random real data sampling with $\delta=\frac{|\mathcal{S}|}{|\mathcal{T}|}$is particularly noteworthy for the privacy of initialization samples. Hence, we propose the KT plugin to protect the privacy of initialization samples.
>     - We implemented a **fix-target membership inference attack [1] on line 472** **(table 3)** specifically to attack the privacy of the initialized data. After using the KT plugin, the initialized sample data can be protected against MIA.
>
> [1] Ye, Jiayuan, et al. "Enhanced membership inference attacks against machine learning models." *Proceedings of the 2022 ACM SIGSAC Conference on Computer and Communications Security*. 2022.

---

> ### Author Response · Authors · 2024-11-22
> **Response to Reviewer rmpC (3/3)**
>
> > W3: The comparison with differentially private generators appears to be unfair.
> >
>
> We appreciate the reviewer's concern about ensuring fair privacy comparisons. We will first explain why the KT plugin is compared with DP-generators, then justify the choice of $\epsilon=10$ for comparison, and finally, we will supplement experiments to create trade-off curves to maintain fairness.
>
> - **KT can achieve comparable privacy protection capabilities to DP-generators while focusing on the usability of the distilled dataset:** In the previous response, we discussed the privacy protection of the entire dataset distillation process and the enhanced privacy protection provided by the KT plugin, which is based on the KL divergence analysis of adjacent datasets. This demonstrates that the distilled dataset possesses properties similar to differential privacy. Although the KT plugin cannot directly provide privacy guarantees, it can be compared to DP-generators in terms of their defense against membership inference attacks. **It also reflects that KT-DATM significantly enhances the usability of data under limited IPC conditions, a capability that existing DP-generators cannot achieve.**
> - $\epsilon=10$ **is already a relatively relaxed condition for DP-generators:** We originally presented results only for $\epsilon=10$ because we emphasize that the information aggregation capability of existing DP-generators is far inferior to that of dataset distillation methods. An $\epsilon$ value of 10 already represents a relatively relaxed privacy protection.
>     - If $\epsilon$ were to be set even smaller, the usability of the data would be significantly worse. **Our KT method achieves the best data information aggregation effect under similar MIA defense and explicit privacy protection.**
> - **Add trade-off curves for fair comparison**: We should present more comprehensive results to readers to facilitate a better understanding of the distinctions between dataset distillation with the KT plugin and DP-generators.  We **have** **created trade-off curves in the appendix for DP-generator methods using different $\epsilon$ values in line 460 on page 9 and the Appendix G (in line 1016 on page 19)**. Notably, for methods like PSG, even as $\epsilon$ approaches infinity, their data utility remains inferior to KT-DATM.

---

> > ### Comment · Reviewer_rmpC · 2024-11-23
> >
> > Thank you for your detailed response.
> >
> > Unfortunately, my concerns have not been adequately addressed.
> >
> > As Reviewer 4Amd also pointed out, without any randomization prior to uniform sampling, your method does not properly satisfy the requirements of differential privacy. Furthermore, the value of $\delta$ assumed in your paper is not appropriate within the context of differential privacy.
> >
> > While you mentioned that $\epsilon=10$ is already a relaxed value, the privacy guarantees of your method appear to deviate significantly even from this relaxed $\epsilon$. In $(\epsilon, \delta)$-differential privacy, relaxing $\delta$ typically introduces a more significant relaxation than $\epsilon$. Therefore, a fair comparison with existing differential privacy methods requires a more rigorous consideration of the corresponding privacy parameters.
> >
> > While I understand that the discussion around differential privacy may be intended to explore how the sampling method can be translated into a differential privacy framework, it lacks the rigor necessary for ensuring compliance with differential privacy guarantees. Aiming to satisfy differential privacy requires a much more careful and thorough discussion.
> >
> > Overall, while your proposed approach is interesting, the discussion and claims regarding differential privacy remain problematic. As such, I will maintain my original negative score.

---

> > > ### Author Response · Authors · 2024-11-23
> > > **Response to Reviewer rmpC**
> > >
> > > Thank you for your additional feedback. We believe there may still be some misunderstanding about our work's privacy claims that we'd like to clarify:
> > >
> > > 1. **Role of Proposition 1**:
> > >     - Proposition 1 serves to rigorously prove the privacy leakage inherent in existing data distillation methods. The large $δ$ value precisely quantifies this leakage risk, aligning with Dwork's “Just a Few.” discussion (at last paragraph on page 9) on partial sample privacy exposure [1].
> > >     - This analysis reveals a critical **Explicit Privacy Leakage** rather than indicating a limitation of our method. Indeed, it is the key motivation driving our development of the KT plugin.
> > >     - The source of randomness in random sampling is the randomized response, as described in the example of coin flipping by Dwork [1], where our initial sample is not specified to a particular sample. Furthermore, our proof fully adheres to the definition of differential privacy. If you believe that the conditions of differential privacy are not met, please point out the error in the proof process.
> > > 2. **Our Complete Privacy Protection Approach**:
> > >     - The privacy protection in our final method stems from the complete pipeline:
> > >         - The KT plugin protecting initialization samples
> > >         - Matching optimization with KL divergence properties
> > >         - Not merely the random sampling analyzed in Proposition 1
> > >     - Our theoretical foundation (Theorem 1 and 2) is based on KL divergence analysis of adjacent datasets, which provides properties distinct from, though analogous to, differential privacy
> > > 3. **Comparison with DP Methods**:
> > > - To be explicit: we do not claim our final method achieves differential privacy. Its effective protection of explicit privacy and the reduction of the distributional difference between adjacent datasets in the paradigm of original dataset distillation.
> > > - Our comparisons focus on:
> > >     - Practical privacy protection (e.g., MIA defense capability)
> > >     - Data utility under comparable privacy protection levels
> > >     - We have created trade-off curves in the appendix for DP-generator methods using different $\epsilon$ values in line 460 on page 9 and the Appendix G (in line 1016 on page 19).
> > >
> > > We need to objectively assess the explicit privacy protection capabilities of KT. Proposition 1 merely serves as background information, and the differential privacy it satisfies would **leak the privacy of the initialization sample**, which is unacceptable. Therefore, we define explicit privacy and the KT plugin. You may also refer to my response to Reviewer 4Amd, which outlines the entire article's narrative logic.
> > >
> > > We believe that Proposition 1 may lead readers to mistakenly consider the differential privacy of random sampling as reliable. We will revise the original content to point out that it exposes the privacy of the initialization sample. Subsequently, we will explain that the second stage of dataset distillation introduces additional random perturbations to the distilled initialization samples.
> > >
> > > [1] Dwork, Cynthia, and Aaron Roth. "The algorithmic foundations of differential privacy." *Foundations and Trends® in Theoretical Computer Science* 9.3–4 (2014): 211-407.

---

> > > > ### Comment · Reviewer_rmpC · 2024-11-23
> > > >
> > > > Thank you for your further response. I acknowledge my earlier misunderstanding and now have a clearer understanding of your approach. Initially, I believed that Proposition 1 aimed to quantify how well the sampling process satisfies differential privacy to justify its privacy protection. However, as clarified in your response, Proposition 1 is instead intended to quantify the privacy leakages in the sampling process. I appreciate this important clarification and sincerely apologize for my earlier misunderstanding.
> > > >
> > > > That said, the organization of the paper still requires refinement. As highlighted in your response, revisions are necessary to make the claims and contributions of this paper clearer and more compelling. I strongly encourage you to address these points in your revision.
> > > >
> > > > I remain highly interested in the proposed explicit privacy approach and, in light of the clarification, will increase my score slightly.
> > > >
> > > > Thank you again for the thoughtful discussion.

---

> ### Author Response · Authors · 2024-11-23
> **Appreciation for Understanding and Article Organization Improvements**
>
> Dear Reviewer rmpC,
>
> Thank you for your thoughtful feedback and for acknowledging the clarification regarding Proposition 1. We greatly appreciate your careful review and the time you've spent understanding our work more deeply.
>
> We have made substantial revisions to improve the clarity of our presentation, particularly focusing on (1) **contributions** in lines 102-112 on page 2 and (2) **statement of Proposition1** in lines 237-252 on page 5.
>
> - Our revised contributions emphasize that we are the first to propose explicit privacy (in lines 102 on page 2), as well as the theoretical analysis of the root cause of the explicit privacy leakage and the weakening defense against membership inference attacks as IPC improves, which is due to the random initialization of training sample sampling (in lines 104 on page 2).
> - Claims on Proposition 1 in lines 237-252 on page 5 now clearly point out that the differential privacy achieved through initialization is unreliable, with $\delta=\frac{|\mathcal{S}|}{|\mathcal{T}|}$ serving as a clear indicator of initial data privacy leakage. This observation directly motivates our introduction of explicit privacy protection.
> - Furthermore, in line 253 on page 5, we emphasize that the perturbations applied to initialized distilled samples in Phase 2 are insufficient for privacy protection. This crucial point is reinforced in Remark 1 (line 280), providing a more coherent narrative of our contributions.
>
> These revisions **(highlighted in blue in the updated PDF)** aim to prevent similar misunderstandings and present our explicit privacy findings more clearly. We appreciate your increased confidence in our work and remain open to any additional suggestions for improvement.
>
> If our revision has addressed your concern, we kindly request you to re-consider your score. We greatly value your assessment and believe our improvements align with your constructive feedback.

---

> > ### Comment · Reviewer_rmpC · 2024-11-27
> >
> > The proposed explicit privacy approach is interesting but exhibits weaknesses in terms of formal privacy guarantees. While the proposed privacy-enhancing technique may be practical in certain scenarios, it carries significant risks when applied more generally. To address these concerns, the inclusion of formal privacy guarantees and more precise, sophisticated discussions is necessary to clarify the novelty and contributions of this work.
> >
> > Additionally, as noted by another reviewer, the paper appears to overclaim in its title and writing. A comprehensive revision is strongly recommended to improve clarity, coherence, and the accuracy of its claims.
> >
> > Given these concerns, I maintain my score as "below the acceptance threshold."

---

> > > ### Author Response · Authors · 2024-11-28
> > > **Response to Reviewer rmpC**
> > >
> > > We would like to express our sincere gratitude for your thorough review and valuable suggestions for improving our paper. Your feedback has been instrumental in helping us refine our work.
> > >
> > > Regarding the explicit privacy approach, we acknowledge that it presents privacy risks in data publishing scenarios. The proposed KT plugin is specifically designed for the dataset distillation task. We have taken your advice to revise the title and writing logic to avoid overstatements and to emphasize our contributions more accurately.
> > >
> > > In our general response, we have provided further explanations on the modifications made throughout the paper and the explicit privacy considerations.
> > >
> > > Thank you once again for your constructive comments. We believe that the revisions we have made will address your concerns and improve the overall quality of our work.

---

> ### Author Response · Authors · 2024-12-02
> **A Kind Reminder to Reviewer rmpC**
>
> Dear Reviewer rmpC,
>
> Thank you once again for your insightful feedback on our submission. We would like to remind you that the discussion period is concluding. To facilitate your review, we have provided a concise summary below, outlining our responses to each of your concerns:
>
> - **Understanding Explicit Privacy**: Differential privacy can protect membership identity, but it does not guarantee the prevention of privacy leakage issues. Therefore, in the context of data release, it is necessary to protect the data itself. Currently, the release of distilled data has exposed the privacy information of initialization samples. The use of the KT plugin in distilled data can simultaneously defend against both explicit privacy leakage and membership inference attacks.
> - **A Comprehensive Revision**: We have revised the title and the main text to avoid over-claiming, added scene images and descriptions to better understand the distillation data scenario, clarified the relationship with differential privacy, and reorganized the contributions of this paper.
>
> More detailed information can be found in the general response and the revised PDF. We are grateful for your insightful comments and are eager to confirm whether our responses have adequately addressed your concerns. We look forward to any additional input you may provide.
>
> Warm regards,
>
> The Authors of Submission 10266.

---

### Official Review · Reviewer_opNC · 2024-11-04

**Soundness:** 3
**Presentation:** 3
**Contribution:** 3
**Rating:** 8
**Confidence:** 4

**Summary:**

The paper addresses significant privacy risks in data distillation, particularly noting that a high number of images per class (IPC) can cause distilled images to closely resemble the original data, leading to substantial privacy leakage. This concern is theoretically supported, demonstrating that higher IPC values in naive distillation increase the risk of data leakage. To mitigate these risks, the authors propose  **Kaleidoscopic Transformation (KT)**, a technique that introduces randomized transformations to real data during the initialization phase. This approach enhances privacy by adding an additional layer of randomness to the distillation process. The effectiveness of KT in strengthening privacy is theoretically validated by  **Theorem 2**. Additionally, the paper introduces a new metric,  **explicit privacy leakage**, defined as the average minimum LPIPS distance between the distilled and original datasets, to more accurately quantify privacy risks.

**Strengths:**

1.  **Well-Motivated Problem:**  The paper validly raises significant concerns about the privacy implications of naive data distillation, showing deep leakage when high IPC conditions exist.

2.  **Effective Randomization with Kaleidoscopic Transformation:**  The proposed KT approach is intuitive and well-motivated, as introducing randomness during initialization adds another layer of privacy protection. This idea is supported by theoretical claims, and  **Theorem 2**  provides a solid foundation demonstrating that KT strengthens privacy in the resultant distilled datasets.

3.  **Introduction of Explicit Privacy Leakage as a Metric:**  Authors introduce  **explicit privacy leakage**—calculated as the average minimum LPIPS distance between the distilled and original datasets—for quantifying privacy risks.

4.  **Experimental Evaluation:**  The authors use multiple datasets and baselines during experimental evaluation of KT

**Weaknesses:**

1.  **Initial Lack of Clarity in Presentation:**  While the paper writing is good , the first two pages were hard to follow. Technical terms such as IPC and explicit privacy leakage are introduced later but the figure is shown earlier and discussion is made in the first two pages without adequate explanation, making the reading the paper challenging. Additionally, Figure 1 presents percentage metrics for differential privacy and explicit privacy leakage but lacks explanations of what these "percentages" represent and how the metrics are calculated.

2.  **Undefined Terms in Theorems:**  Variables like  P  and  Q  in Theorems 1 and 2 are not clearly defined. Although looking at the proof helps, providing explicit definitions would enhance clarity and understanding.

3.  **Ambiguity in Figure 4:**  The meaning of the numbers presented in Figure 4 is unclear, necessitating further explanation to interpret the results effectively.

4.  **Claim :  "Free" Privacy Improvement is not substantiated:**  The claim that KT provides a “free improvement in privacy without significant computational overhead" seems inaccurate (Line 461). For instance, Figure 5 shows a trade-off between privacy and utility":  increased transformations lead to reduced utility. While it is possible to claim that KT may offer an improved balance under certain conditions, describing it as "free" implies no utility loss, which is misleading with Figure 5 results.

5.  **Claim: "Improved balanced performance with Baselines" needs a little more explanation:**  In Table 2, the paper shows certain trade-off points between data generator methods for a fixed privacy budget of 10. It is unclear whether these baseline methods depend on a fixed budget or if comparisons can be made with other epsilon values. It would be ideal if a tradeoff curve could be shown that shows KT offfers better tradeoff. Looking at Table 2 I cant be sure of this conclusion as I can only see KT has high utility. This ambiguity makes it difficult to substantiate the claim that "our approach demonstrates a balanced performance in privacy preservation and data utility"

**Questions:**

Look at weaknesses section for my concerns.

---

> ### Author Response · Authors · 2024-11-22
> **Response to Reviewer opNC**
>
> We thank the reviewer for the comments and questions! Please find our responses to your raised questions below:
>
> > W1&W2&W3: Initial Lack of Clarity in Presentation, Undefined Terms in Theorems and Ambiguity in Figure 4.
> >
>
> We agree that improving the readability of our paper, especially in the early sections, would make our contributions more accessible to readers. We have made the following specific revisions to address these concerns.
>
> - In Figure 1, we now provide explanations for all technical terms (IPC, explicit privacy leakage, MIA, and LPIPS) in the caption, with clear references to where their formal definitions appear in the paper.
> - We have added explicit definitions for all variables used in Theorem 1 to ensure mathematical rigor and readability.
> - We have enhanced Figure 4's caption by explaining the numerical values and clearly distinguishing between the orange and green regions for better interpretation of the results.
>
> > W4: Claim : "Free" Privacy Improvement is not substantiated.
> >
>
> We appreciate the reviewer's careful observation regarding our use of the term "free." We would like to clarify our statement as follows:
>
> - The primary objective of our KT plugin is to maintain the utility of distilled datasets while providing **explicit privacy protection**. In the context of dataset distillation as our main focus, KT additionally offers enhanced defense against membership inference attacks.
> - More specifically, regarding computational cost, we would like to highlight that the introduction of our KT plugin does not incur **additional computational overhead** in the distillation process.
>     - **Additional computational overhead of DP-generator**: In contrast, existing DP-generator approaches typically require noise injection operations at each iteration step [1] and some methods [2, 3] rely on pre-training models with public data, both of which introduce substantial computational burden.
>     - **Comparing additional time through experiment:** The KT plugin enhances the distillation samples three times during the initialization phase and averages them, completing the experiment with IPC=50 on TinyImageNet in just **3 seconds**. In contrast, PSG increases the computational cost by **7 hours** when $\epsilon=10$ compared to without noise.
>     - Our method demonstrates superior generation efficiency by avoiding these extra computational costs while defending against **membership inference attacks and explicit privacy leakage**.
>
> [1] Chen, Dingfan, Raouf Kerkouche, and Mario Fritz. "Private set generation with discriminative information." *Advances in Neural Information Processing Systems* 35 (2022): 14678-14690.
>
> [2] Wang, Haichen, et al. "dp-promise: Differentially Private Diffusion Probabilistic Models for Image Synthesis." USENIX, 2024.
>
> [3] Li, Kecen, et al. "{PrivImage}: Differentially Private Synthetic Image Generation using Diffusion Models with {Semantic-Aware} Pretraining." *33rd USENIX Security Symposium (USENIX Security 24)*. 2024.
>
> > W5: Claim: "Improved balanced performance with Baselines" needs a little more explanation.
> >
>
> We appreciate the reviewer's valuable feedback regarding the comparative analysis of privacy-utility trade-offs. We would like to clarify that the DP-generator baselines in our paper indeed **support adjustable privacy budgets (**$\epsilon$ **values)**.
>
> Our initial presentation at $\epsilon=10$ was designed to demonstrate that our KT method provides explicit privacy protection for dataset distillation while maintaining data utility. The comparison with DP-generators was intended to demonstrate that our KT plugin retains the capability to effectively distill information from the original dataset. While DP-generators provide differential privacy guarantees, they exhibit inferior downstream task accuracy with the same IPC, even when achieving comparable MIA attack success rates.
>
> To ensure comprehensive and fair comparison, we acknowledge the importance of examining performance across different privacy budgets. Following your suggestion, we  have **added trade-off curves in line 460 on page 9 and the Appendix G (in line 1016 on page 19)** to illustrate how DP-generators perform in terms of MIA defense and data utility under varying $\epsilon$ values. **Notably, for methods like PSG, even as $\epsilon$ approaches infinity, their data utility remains inferior to KT-DATM.**

---

> > ### Comment · Reviewer_opNC · 2024-11-23
> >
> > Thanks for this.  I appreciate the response.  My main concerns were 1: Presentation of the paper especially technical terms 2: "Free" privacy claim 3: Comparison with baseline. The authors promise to  address 1) and 3) in the revised version. Regarding 2) while I appreciate that "free" means that reduced computational overhead and not utility loss I would advise authors to be circumspect in the claims and use of terminology.  Also, I would still maintain that the writing should acknowledge the limitation in that KT provides better tradeoffs and not free privacy as some previous works have incorrectly used the term "free". I think that would position the paper in a more reasonable light.
> >
> > I did not find anything wrong in the technical aspects of the paper and only circumspect writing remained by biggest concern, I would increase the score.

---

> ### Author Response · Authors · 2024-11-23
> **Thank you for the very positive review and rating!**
>
> Dear Reviewer opNC,
>
> We sincerely thank you for your thoughtful reply and are pleased that our response has addressed most of your concerns, as reflected in the increased rating (8).
>
> Your feedback about being more circumspect with terminology, particularly regarding the "free privacy" claim, remains especially valuable. We acknowledge that this aspect requires further refinement and will revise our language throughout the paper to more accurately characterize our method's benefits.
>
> Once again, we deeply appreciate your thoughtful and constructive feedback!
>
> Best regards,
>
> Authors

---

### Author Response · Authors · 2024-11-22
**General Response**

We sincerely thank the reviewers for their insightful feedback. We are delighted that the reviewers acknowledged the **clarity** and **novelty** of our motivation (Reviewers opNC, rmpC), the **significance** of our findings (Reviewers opNC, rmpC), the solid theoretical foundation of our method (Reviewer opNC), the extensive and well-designed **experiments** (Reviewers opNC, rmpC).

Furthermore, our contributions to identifying **a novel privacy leakage risk** in dataset distillation, proposing the innovative Kaleidoscopic Transformation (KT) module, introducing the **explicit privacy leakage** metric, and establishing a comprehensive evaluation framework are highly appreciated. The **KT module** is recognized as a creative and intuitive approach (Reviewer 4Amd), supported by Theorem 2, which demonstrates its effectiveness in enhancing privacy protection (Reviewer opNC). Our experimental validation, covering multiple datasets and baselines, further solidifies the efficacy of our method in balancing utility and privacy.

Additionally, the **clear and concise presentation** of our work, facilitating reproducibility and understanding, was also acknowledged (Reviewers rmpC, 4Amd, 6CFr)

### Common Question about Proposition 1

We assume that some reviewers may have misunderstandings about Proposition 1, and we would like to clarify this as follows. We argue that **Proposition 1 motivated us to focus on explicit privacy risks and led us to propose the KT plugin**. At the same time, data distillation is a two-stage process, and we need to comprehensively consider privacy protection in the distilled dataset.

- **Random sampling initialization is not a perfect privacy protection scheme, but it is of great significance**:
    - By randomly selecting samples, the privacy of most training data is effectively protected.
    - It reveals the potential privacy leakage risks in the initialization phase.
- **A larger δ value (δ=|S|/|T|) implies**: There is a risk of privacy leakage in the initialization samples, which has inspired us to pay attention to the privacy of initialization training samples, thereby proposing explicit privacy and the KT plugin.
- **The overall framework is a two-stage privacy protection mechanism**:
    - Stage 1: Random sampling initialization.
    - Stage 2: By introducing perturbations to the initialization samples through a matching optimization process, we analyze both the normal data distillation process and the process after introducing the KT plugin (Theorem 1 and 2), and believe that they possess properties similar to differential privacy.
- Regarding our focus on the privacy of initialization samples, we also conducted experiments on **fix-target membership inference attacks**, reducing the privacy attack success rate of initialization samples from 99.5% to 54.1%, as detailed in Appendix H on page 20.

---

### Author Response · Authors · 2024-11-28
**General Response**

We would like to express our sincere gratitude for the detailed reviews and constructive feedback provided by the reviewers. We have thoroughly revised our manuscript to address the concerns regarding overclaiming and the importance of explicit privacy, **with the changes highlighted in teal for clarity**. Additionally, we believe there may have been some misunderstandings regarding explicit privacy, which we will clarify in this response.

**Avoid Overclaiming and Enhance Readability**

We have corrected the title and revised the entire manuscript to ensure that the context of dataset distillation is clearly understood, avoiding any exaggerated statements.

- **Revised Title:** "Privacy Leakage Prevention in Distilled Datasets: Transforming Initial Private Data"
This new title emphasizes the issue of privacy leakage in distilled datasets, particularly focusing on the initialization of private data.
- **Revised Introduction:**
    - We have reinforced the primary objective of dataset distillation and highlighted the new privacy challenges in the data-release scenarios (add Figure 1).
    - We have also reduced the association between dataset distillation and differential privacy. Dataset distillation does not provide precise differential privacy but rather analyzes distilled data through adjacent datasets to measure the limited the individual samples privacy on the distillation results, thereby effectively defending against membership inference attacks.

**Clarify Misunderstandings on Explicit Privacy**

We believe there may have been a misunderstanding regarding explicit privacy, which we would like to clarify. Explicit privacy is crucial in data-release scenarios:

- **Differential Privacy Limitations:** Differential privacy protects whether individual data is involved but does not protect against data leakage. In Dwork's book [1], on page 22, under the section "**What differential privacy does not promise**," it states: "In particular, **if the survey teaches us that specific private attributes correlate strongly with publicly observable attributes, this is not a violation of differential privacy**, since this same correlation would be observed with almost the same probability independent of the presence or absence of any respondent.” Therefore, while differential privacy can effectively defend against membership inference attacks, the risk of data leakage remains.
    - Using Reviewer 6CFr's simple example of differential privacy, mapping the input to a fixed value can also achieve DP. However, if the fixed value is private information, it does not violate the rules of differential privacy, but an attacker can still obtain private information from it. ***Therefore, explicit privacy measurement is necessary.***
- **Explicit Privacy is a unique and already exposed risk in data-release scenario:** Explicit privacy is essential as it addresses a unique privacy risk in data-release scenarios. Unlike privacy-preserving machine learning models, which release models rather than data, distilled datasets directly release distilled data. Attackers can directly hold this data for visualization or model training, necessitating the emphasis on data leakage risks. Current state-of-the-art dataset distillation algorithms like DATM **already directly leak private data used for initialization** under high IPC conditions. We need to focus on explicit privacy leakage and provide protection.
- **Integration with Membership Inference Attacks:** Explicit privacy **is not an isolated privacy evaluation metric**. We evaluate the privacy protection of released data or distilled datasets through both membership inference attacks and explicit privacy. In our experiments, we conducted both membership inference attacks and data visualization. Our dataset distillation method, combined with the KT plugin, ***effectively addresses explicit privacy leakage and further enhances the ability to resist membership inference attacks***, achieving the best trade-off between privacy and utility.

We appreciate the time and effort all reviewers have invested in reviewing our work and providing valuable suggestions. We hope that the revised version addresses the concerns raised and contributes to the development of privacy protection in the dataset distillation community.

[1] Dwork, Cynthia, and Aaron Roth. "The algorithmic foundations of differential privacy." *Foundations and Trends® in Theoretical Computer Science* 9.3–4 (2014): 211-407.

---

### Meta-Review · Area_Chair_uAbD · 2024-12-22

**Metareview:**

The paper addresses significant privacy risks in data distillation, particularly noting that a high number of images per class (IPC) can cause distilled images to closely resemble the original data, leading to substantial privacy leakage. This concern is theoretically supported, demonstrating that higher IPC values in naive distillation increase the risk of data leakage. To mitigate these risks, the authors propose Kaleidoscopic Transformation (KT), a technique that introduces randomized transformations to real data during the initialization phase. This approach enhances privacy by adding a layer of randomness to the distillation process. The effectiveness of KT in strengthening privacy is theoretically validated by Theorem 2. Additionally, the paper introduces a new metric, explicit privacy leakage, defined as the average minimum LPIPS distance between the distilled and original datasets, to more accurately quantify privacy risks.


Although the reviewers diligently responded to the reviewers' comments, the consensus is that this paper still needs more work to improve clarity, coherence, and the accuracy of its claims.

**Additional Comments On Reviewer Discussion:**

Several reviewers think this paper still needs more work to improve, in terms of the presentation of their work.

---

### Decision · Program_Chairs · 2025-01-22

Reject